# The First Transcriptomic Atlas of the Adult Lacrimal Gland Reveals Epithelial Complexity and Identifies Novel Progenitor Cells in Mice

**DOI:** 10.3390/cells12101435

**Published:** 2023-05-21

**Authors:** Vanessa Delcroix, Olivier Mauduit, Hyun Soo Lee, Anastasiia Ivanova, Takeshi Umazume, Sarah M. Knox, Cintia S. de Paiva, Darlene A. Dartt, Helen P. Makarenkova

**Affiliations:** 1Department of Molecular and Experimental Medicine, Scripps Research Institute, La Jolla, CA 92037, USA; vdelcroix@scripps.edu (V.D.); hslee@scripps.edu (H.S.L.); aivanova@scripps.edu (A.I.); umazume@scripps.edu (T.U.); 2Department of Ophthalmology, College of Medicine, The Catholic University of Korea, Seoul 06591, Republic of Korea; 3Department of Cell and Tissue Biology, University of California San Francisco, San Francisco, CA 94143, USA; sarah.knox@ucsf.edu; 4Program in Craniofacial Biology, University of California San Francisco, San Francisco, CA 94143, USA; 5The Ocular Surface Center, Department of Ophthalmology, Cullen Eye Institute, Baylor College of Medicine, Houston, TX 77030, USA; cintiadp@bcm.edu; 6Schepens Eye Research Institute of Massachusetts Eye and Ear, Harvard Medical School, Boston, MA 02114, USA; darlene_dartt@meei.harvard.edu

**Keywords:** lacrimal gland, scRNAseq, progenitors, sex dimorphism, exocrine gland

## Abstract

The lacrimal gland (LG) secretes aqueous tears. Previous studies have provided insights into the cell lineage relationships during tissue morphogenesis. However, little is known about the cell types composing the adult LG and their progenitors. Using scRNAseq, we established the first comprehensive cell atlas of the adult mouse LG to investigate the cell hierarchy, its secretory repertoire, and the sex differences. Our analysis uncovered the complexity of the stromal landscape. Epithelium subclustering revealed myoepithelial cells, acinar subsets, and two novel acinar subpopulations: *Tfrc^hi^* and *Car6^hi^* cells. The ductal compartment contained *Wfdc2+* multilayered ducts and an *Ltf+* cluster formed by luminal and intercalated duct cells. *Kit+* progenitors were identified as: *Krt14+* basal ductal cells, *Aldh1a1+* cells of *Ltf+* ducts, and *Sox10*+ cells of the *Car6^hi^* acinar and *Ltf+* epithelial clusters. Lineage tracing experiments revealed that the *Sox10*+ adult populations contribute to the myoepithelial, acinar, and ductal lineages. Using scRNAseq data, we found that the postnatally developing LG epithelium harbored key features of putative adult progenitors. Finally, we showed that acinar cells produce most of the sex-biased lipocalins and secretoglobins detected in mouse tears. Our study provides a wealth of new data on LG maintenance and identifies the cellular origin of sex-biased tear components.

## 1. Introduction

The lacrimal gland (LG) is a tubuloacinar organ that produces the aqueous component of the tear film, which is essential for maintaining and protecting the cornea. The LG is primarily composed of three major epithelial cell types: ductal cells (luminal and basal subtypes), serous acinar cells expressing basic helix–loop–helix family member a15 (BHLHA15 or MIST1), and myoepithelial cells (MECs) labeled by aortic smooth muscle actin (ACTA2, also known as α-smooth muscle actin (α-SMA)). Additionally, the stromal compartment of the LG contains blood vessels, fibroblasts, multiple immune cell types, and tear-stimulating nerves. Recent studies have suggested that subtypes of LG acinar cells may also produce mucins [1,2,3]; therefore, the cell types constituting the LG, as well as their functions, may not be limited to just a watery secretion.

Several studies have used genetic cell lineage tracing analyses to investigate the development and differentiation of cells in the LG [4,5,6]. Our research indicates that the embryonic multipotent progenitor cells that express keratins 5 and 14 (*Krt5*, *Krt14*) and runt-related transcription factor 1 (*Runx1*) differentiate mainly into the proximal SRY-box transcription factor 10 (SOX10)-/SOX9+ lineage, which produces ductal cells, and into the distal SOX10+/SOX9+ lineage, which forms epithelial endbuds. These distal buds give rise to two lineages: the SOX10+/ACTA2+ MEC lineage and the SOX10+/ACTA2- acinar cell lineage [5], which further differentiates into SOX10-/BHLHA15+ acinar cells [4]. Recent work from Athwal et al. [6] showed that in the LG and several other exocrine glands, the distal part of the duct system is also derived from embryonic *Sox10+* multipotent progenitors. In the postnatal LG, lineage restriction occurs, which limits the ability of lineage-specific cells to generate different cell types in resting conditions. However, some cell types, such as MECs, retain a high level of plasticity and can replace damaged cells upon gland injury, contributing to tissue repair [5]. In addition, KIT proto-oncogene receptor tyrosine kinase (KIT)+ mouse epithelial cells may regenerate acinar and ductal cells in vivo [7]. These studies suggest that lineage restriction plays a crucial role in maintaining the identity and function of the gland, while cellular plasticity is involved in its repair and regeneration. In spite of their limitations, these lineage tracing studies have shed light on the complex cellular processes involved in the development and maintenance of the LG, whilst identifying the mechanisms universally shared by exocrine glands.

Single-cell RNA sequencing (scRNAseq) is a valuable technique for identifying novel or poorly studied cell types and predicting their functions in an organ. The first scRNAseq analysis of mouse LG in the embryonic stage (E.16) and early postnatal development (P4) was performed by the group of Sarah Knox several years ago [4]. In a more recent publication, researchers investigated the human LG using organoids and scRNAseq [8]. However, the small number of cells sequenced and the developmental stages studied [4,8] did not provide a detailed identification of all the cell types present in the LG. Although multicenter initiatives such as the *Tabula Muris* consortium have generated single-cell atlases of various mouse tissues [9], the LG is unfortunately absent from these projects. A deep comprehensive characterization of the adult LG is however critically needed to better understand the pathological conditions associated with LG dysfunction, such as dry eye, as well as the origin of the sex differences observed in its prevalence and its symptoms [10].

Previous publications have indeed suggested that there are sex-related differences in the LGs of both rodents and humans [11,12,13]. However, these differences have been linked primarily to functional factors such as the secretory activity and susceptibility to disease [14,15] and have not yet been thoroughly investigated at the single-cell level. Stopkova et al. [14] and Karn et al. [16] analyzed sexual dimorphism in mouse tears and found differences between males and females in the abundance of bactericidal proteins, exocrine-gland-secreted peptides, and secretoglobins. These studies suggest that tears and likely the epithelial cells that produce them may have sex-specific signaling capabilities. However, the cell subtypes responsible for sexually dimorphic tear proteins are yet to be identified.

In this study, we utilized scRNAseq to investigate the cellular complexity of the adult mouse LG and to assess sexual dimorphism in relation to tear composition. We established a detailed and representative cell atlas of the LG that enabled us to characterize in depth the composition of the LG stroma, epithelium, and putative epithelial progenitor cells, as well as to discover new specific markers. Finally, we also determined the respective contribution of epithelial subtypes in the composition and sexual dimorphism of mouse tears.

## 2. Materials and Methods

### 2.1. Animals

Mice were housed under standard conditions of temperature and humidity, with a 12 h light/dark cycle and free access to food and water. All experiments were performed in compliance with the ARVO Statement for the Use of Animals in Ophthalmic and Vision Research and the Guidelines for the Care and Use of Laboratory Animals published by the US and National Institutes of Health (NIH Publication No. 85-23, revised 1996) and were preapproved by TSRI Animal Care and Use Committee.

For scRNAseq experiments, 2-month-old C57BL/6J mice were obtained from the Scripps breeding colony. For immunostaining and lineage tracing studies, the following mouse lines were purchased from the Jackson Laboratory (Bar Harbor, ME, USA): MacGreen (RRID:IMSR_JAX:018549), *Pdgfra*^EGFP^ (RRID:IMSR_JAX:007669), and *Sox10*-iCre^ERT2^ (RRID:IMSR_JAX:027651) that were crossed with Ai9 (RRID:IMSR_JAX:007909) to generate the *Sox10*-iCre^ERT2^:tdTOM^fl^ progeny used in this study. The *Acta2*^GFP^ reporter mouse was a kind gift from Dr. Ivo Kalajzic [17].

### 2.2. Single-Cell Isolation from LG

Single-cell suspensions were prepared from male and female LGs. For each sample, six dissected LGs of three animals aged 2 months were combined and processed as follows. First, the LGs were quickly minced on ice in a dish prefilled with cold DPBS (#14190-144, Gibco, Thermo Fisher Scientific, Waltham, MA, USA), rinsed, and transferred to a 12-well plate placed in a shaking water bath (37 °C, 80 rpm) containing 2 mL of digestion medium of following composition: DMEM/F12 (#DF042B, Sigma-Aldrich, Inc., St. Louis, MO, USA), 0.5% BSA (#A4919, Sigma-Aldrich, Inc., St. Louis, MO, USA), 15 mM HEPES (#MT25060CI, Thermo Fisher Scientific, Waltham, MA, USA), 3 mM CaCl_2_ (#501035464, Thermo Fisher Scientific, Waltham, MA, USA), 125 U/mL collagenase IV (#C5138, Sigma-Aldrich, Inc., St. Louis, MO, USA), 1 U/mL dispase II (#D4693, Sigma-Aldrich, Inc., St. Louis, MO, USA), and 20 U/mL DNase-I (#DN25, Sigma-Aldrich, Inc., St. Louis, MO, USA). After 30 min, the samples were gently triturated by pipetting up and down 15 times with a wide-bored 1 mL tip. At 60 and 90 min, the samples were gently triturated 10 to 15 times with a regular 1 mL pipet tip.

At the end of the digestion (90 min in total), the cells were pelleted (300× *g*, 5 min, 4 °C), washed with cold HBSS buffer (#MT21022CV, Thermo Fisher Scientific, Waltham, MA, USA), supplemented with 10 mM HEPES and 1 mM EDTA (#AM9260G, Thermo Fisher Scientific, Waltham, MA, USA), and gently resuspended in 1 mL of accutase solution (#A6964, Sigma-Aldrich, Inc., St. Louis, MO, USA). After 5 min at 37 °C, the cell suspension was pipetted up and down 5 times and left at RT for 2 min. Then, the accutase was inactivated by the addition of 1 mL of DMEM/F12 supplemented with 10% FBS and 15 mM HEPES. The cells were pelleted (300× *g*, 5 min, 4 °C) and incubated for 10 min at RT with HBSS supplemented with 10 mM HEPES, 2 mM MgCl_2_, 2 mM CaCl_2_, and 200 U/mL DNase-I. The cell suspension was sequentially passed 2 times through a 3 mL syringe with needles: 18G, 22G, and 25G. The dissociated cells were filtered through a pre-wet 70-µm mesh cell strainer (#130-098-462, Miltenyi Biotec, Bergisch Gladbach, Germany) and pelleted (300× *g*, 5 min, 4 °C) before incubation for 5 min at RT with 5 mL of RBC lysis buffer 1× (#5831-100, BioVision-Abcam, Waltham, MA, USA) to deplete the erythrocytes. After centrifugation (300× *g*, 5 min, 4 °C) and resuspension into cold HBSS supplemented with 0.5% BSA and 10 mM HEPES, the cells were passed through a pre-wet 30-µm mesh cell strainer (#130-098-458, Miltenyi Biotec, Bergisch Gladbach, Germany).

After centrifugation (400× *g*, 5 min, 4 °C), the dead cells and debris were depleted using the MS columns (#130-042-201, Miltenyi Biotec, Bergisch Gladbach, Germany) and the dead cell removal kit (#130-090-101, Miltenyi Biotec, Bergisch Gladbach, Germany) according to the manufacturer’s instructions. The cell suspension was washed 3 times with cold scRNA-seq buffer composed of DPBS (#14190250, Thermo Fisher Scientific, Waltham, MA, USA), supplemented with 0.04% BSA, and pelleted at 400× *g* for 5 min at 4 °C. The number of cells and the viability rate (78–79%) were evaluated using trypan blue (#1691049, Thermo Fisher Scientific, Waltham, MA, USA) staining analyzed with the Countess 3 (Thermo Fisher Scientific, Waltham, MA, USA). The cell suspension volumes were then adjusted with cold scRNAseq buffer to obtain between 700 and 1200 cells per µL, and the cells were counted again to obtain the final cell density. The cells were kept on ice at all times.

### 2.3. Single-Cell RNA Sequencing (scRNAseq)

Female samples were processed at the Scripps Genomics core (La Jolla, CA, USA). Male samples were prepared on a different day by the Next Generation Sequencing (NGS) Core at the Salk institute (La Jolla, CA, USA) in similar conditions. For each sample (n = 3 mice), 15,000 cells were processed for scRNAseq using the Chromium Single Cell 3′ kit (v3.1 Chemistry) from 10× Genomics (Pleasanton, CA, USA) according to the manufacturer’s protocols. The cDNA was amplified by 11 cycles. Analysis of the cDNA and libraries was conducted using Bioanalyzer or TapeStation (Agilent, Santa Clara, CA, USA). The libraries were sequenced multiple times by Nextseq 550 systems and Novaseq 6000 sequencer (Illumina, San Diego, CA, USA) to reach the targeted minimal sequencing depth (20,000 reads/cell).

### 2.4. Sequencing Data Processing and Analysis

The demultiplexing, read alignment, and identification of empty droplets were conducted with CellRanger v6.1.2 software (10× Genomics, Pleasanton, CA, USA). The resulting filtered feature-barcode matrices were analyzed on RStudio (R v4.1.1). One female sample (3 female mice in total) and two male samples (6 male mice in total) were retained for further analysis in this study. The datasets were deposited in the Gene Expression Omnibus (GEO) database (www.ncbi.nlm.nih.gov/geo, accessed on 15 May 2023) under accession #GSE232146. All samples displayed a relatively low fraction of reads assigned to cells (<70%), suggesting a high ambient RNA level.

To reduce the ambient RNA contamination in the individual cells and identify doublets, we used the R package SingleCellTK (v2.2.0) [18] (Appendix A). Briefly, using Seurat (v4.1.1)/SCTransform (v0.3.5) [19,20], we first identified the broad cell types (8–9 clusters) by analyzing the filtered matrix generated by CellRanger that contains cell-associated barcodes (including low-quality/apoptotic cells). Then, the filtered matrix and cell identities were subjected to analysis with SingleCellTK that included celda/DecontX (v1.12.0, to reduce cross-contamination [21]) and multiple methods to identify doublets (scDblFinder [22], DoubletFinder [23], and scds_hybrid [24]). Because any doublet detection method may identify false positives, we chose to classify as ‘true’ doublets the cells displaying a relatively high score (i.e., the probability) with two different methods: DoubletFinder and scDblFinder (that integrates co-expression scores as predictors, similar to scds). The relevant score thresholds for high-quality cells were determined by joint inspection of the scores obtained by the different methods and by studying the gene expression following high-resolution clustering. For all samples, we thus excluded cells with a contamination score >0.7 and those we considered as ‘true’ doublets if they had scores >0.25 with both DoubletFinder and scDblFinder. The major source of cross contamination was the acinar population, as shown by the many transcripts for lipocalins and secretoglobins (e.g., *Mucl2*, *Obp1a*, *Scgb2b2*, etc.) being ectopically detected in the entire dataset and highly corrected by DecontX (Appendix A). Due to their secretory function, mature acinar cells have a large RNA content, and any alteration in the membrane permeability due to enzymatic and mechanical dissociation may lead to a non-neglectable amount of RNA being released into the cell suspension.

Finally, only cells meeting all the last quality criteria (nUMI > 500, nGene > 200, proportion of mitochondrial genes <15%) were retained for the analysis presented in this study (7056 cells for the females and 4723 and 4493 cells for each male sample, respectively) (Appendix A). Whilst most studies use a 5% cutoff for the proportion of mitochondrial transcripts [25], we found that doing so would aberrantly filter out specific epithelial subpopulations otherwise satisfying quality thresholds and showing relevant expression profiles. To verify that only high-quality cells remained, we performed high-resolution clustering using decontaminated gene counts to investigate cluster markers and the expression of canonical marker genes in each sample independently.

For whole LG analysis, the normalization and scaling of the DecontX-corrected counts were conducted with the Seurat/SCTransform package. The integration of samples was conducted using canonical correlation analysis (CCA), following Seurat’s recommended analysis pipeline. The cells were clustered using Seurat’s graph-based clustering approach, and the dimension reduction was performed using Uniform Manifold Approximation and Projection (UMAP). The identification of subclusters in the whole LG dataset was conducted using the *FindSubCluster* function from Seurat. For epithelial subclustering, barcodes that belonged to the epithelial compartment were subset and subjected to a separate analysis similarly. To study the gene expression, the DecontX-corrected counts were normalized using the *NormalizeData* function from Seurat.

Data from the LG at postnatal day 4 (P4, deposited under GEO accession #GSM2671416) were downloaded from PanglaoDB (panglaodb.se) and analyzed with R/Seurat using SCTransform normalization and scaling for cell clustering. The relevant groups of cells were evidenced by using the *FindSubCluster* function.

All the R scripts and Seurat objects used in this study can be found on Zenodo (zenodo.org, accessed on 15 May 2023) with https://doi.org/10.5281/zenodo.7927055, accessed on 15 May 2023.

### 2.5. Cell-Lineage Tracing Experiments and Immunostaining

Tamoxifen (TM, 1 g) (#T5648-1G, Sigma-Aldrich, Inc., St. Louis, MO, USA) was resuspended in 2 mL ethanol and diluted in filtered corn oil at 20 mg/mL. The solution was vortexed and placed on a shaker at 45 °C overnight until fully dissolved. TM-inducible Cre-recombinase was activated in mice aged 1.5–2 months old by intraperitoneal injection (0.1 mg TM per g of weight). The noninjected littermates were analyzed to ensure the absence of spontaneous Cre-mediated recombination in the LG.

All the immunostaining experiments were performed on separate days in at least 4 mice between 1 and 6 months old, as indicated in figure legends. Both males and females were investigated. The mice were sacrificed by CO_2_ inhalation followed by cervical dislocation. The harvested LGs were fixed with 2% PFA in PBS (pH 7.4) for 45 min and processed for whole-mount immunostaining or frozen in 2-methylbutane (Sigma-Aldrich, Inc., St. Louis, MO, USA) cooled by liquid nitrogen. Frozen sections (15 μm) were cut with a Hacker/Bright OTF5000-LS004 Cryostat (Hacker Instruments and Industries Inc., Winnsboro, SC, USA) and blocked with 5% BSA in Tris-buffered saline containing 0.05% Tween 20 (TBST). For the whole mounts, the lobules were permeabilized for 30 min with TBST.

Incubation with the appropriate primary antibodies was performed at 4 °C overnight: ACTA2 (α-SMA) mouse monoclonal antibody (#A2547, RRID:AB_476701, Sigma-Aldrich, Inc., St. Louis, MO, USA), PECAM-1 rat monoclonal antibody (#553370, RRID:AB_394816, BD Biosciences, Franklin Lakes, NJ, USA), E-Cadherin (E-CAD, encoded by *Cdh1*) mouse monoclonal antibody (#610182, RRID:AB_397581, BD Biosciences, Franklin Lakes, NJ, USA), Aquaporin5 rabbit antibody (AQP5, #ab92320, RRID:AB_2049171, Abcam, Cambridge, UK), SOX10 rabbit monoclonal antibody (#ab245760, Abcam, Cambridge, UK), HSP60 rat monoclonal antibody (#66041-1-Ig, RRID:AB_11041709, Proteintech, Rosemont, IL, USA), heparan sulfate proteoglycan rat monoclonal antibody (HSPG2, #MAB1948P, RRID:AB_10615958, Sigma-Aldrich, Inc., St. Louis, MO, USA), PAX6 rabbit polyclonal antibody (#Poly19013, BioLegend, San Diego, CA, USA), KRT14 rabbit monoclonal antibody (#SAB5500124, Sigma-Aldrich, Inc., St. Louis, MO, USA), Neurofilament 200 (NF200) mouse monoclonal antibody (#N5389, RRID:AB_260781, Sigma-Aldrich, Inc., St. Louis, MO, USA), TOM20 mouse monoclonal antibody (#sc-17764, RRID:AB_628381, Santa Cruz Biotechnology, Inc, Dallas, TX, USA), and CD3 rabbit monoclonal (Cat# NB600-1441, RRID:AB_789102, Novus Biologicals/Bio-Techne, Minneapolis, MN, USA). The samples were washed with TBST and then incubated for 45 min at room temperature with the appropriate secondary antibody purchased from Thermo Fisher Scientific (Molecular Probes, Waltham, MA, USA). Nuclei were stained with DAPI. The slides were mounted with Fluoromount-G (#0100-01, SouthernBiotech, Birmingham, AL, USA). To verify that the fluorescent signal was solely attributed to the binding of the primary antibody, we used negative controls in which the incubation step with the primary antibody was intentionally omitted. Images were taken using a Zeiss LSM 880 laser scanning confocal microscope at the Microscopy core of the Scripps Research Institute.

### 2.6. Preparation of LG Sections Stained with Toluidine Blue

The LGs were first fixed in a solution of 3% glutaraldehyde (#01909-10, Polysciences, Warrington, PA, USA) and then washed in a 1 M sodium phosphate buffer (PBS) with a pH of 7.3. They were subsequently post-fixed in an osmium tetroxide (OsO4; #0972A-20, Polysciences, Warrington, PA, USA) solution for one hour. After fixation, the samples underwent dehydration through a series of graded alcohols, acetone, and Polybed 812 plastic resin (#08792-1, Polysciences, Warrington, PA, USA). Finally, they were embedded in plastic block molds using 100% Polybed 812. One-micron sections were obtained using a Leica EMCU ultra microtome and placed on glass slides. The sections were then stained with toluidine blue (#01234-25, Polysciences, Warrington, PA, USA).

### 2.7. Flow Cytometry Analysis of LG Immune Cells

Single-cell suspensions of the LGs from 3–4-month-old females were prepared as previously reported [26]. One million cells were blocked with CD16/CD32 Fc blocker (#553141, BD Biosciences, Franklin Lakes, NJ, USA), washed, incubated with live/dead cell discriminator (IR, #L34994, Molecular Probes/Thermo Fisher Scientific, Waltham, MA, USA), and stained using different antibodies/panels: *panel 1*: PTPRC (CD45) (BV510, clone 30F11, #103138, Biolegend, San Diego, CA, USA) and gamma/delta (γδ) TCR (PE-Cy5, clone GL-3, #15-5711-81, eBioscience™/ThermoFisher, Waltham, MA, USA); *panel 2*: PTPRC (BV510, clone 30F11, #103138, Biolegend, San Diego, CA, USA), ITGAX (CD11c) (FITC, Clone HL3, #557400, BD Pharmingen, San Diego, CA, USA), MHC II (I-A/I-E, encoded by *H2-Ab1*) (PE, clone M5/114.15.2, #557000, BD Pharmingen, San Diego, CA, USA), and ITGAM (CD11b) (PE-Cy7, clone M1/70, #552850, BD Pharmingen, San Diego, CA, USA). The cells were then washed two times and resuspended in FACS buffer and kept on ice until flow cytometry was acquired.

For both panels, the live PTPRC^+^ cells were gated by excluding the live/dead dye, followed by two sequential single-cell gates. The γδ TCR+ cells were gated vs. forward scatter area (FSC-A). In a second panel, the MHC II^+^ cells were gated, followed by an ITGAX vs. ITGAM plot.

A BD LSRII Benchtop cytometer was used for data acquisition, and the data were analyzed using BD Diva Software (BD Pharmingen, San Diego, CA, USA) and FlowJo software (version 10.1; Tree Star, Inc., Ashland, OR, USA). The data shown are representative of three independent experiments.

### 2.8. Comparison with the Tear Proteome

To compare the expression level of the tear components in our scRNAseq data to tear composition, we used the tear proteome data previously published by Stopkova et al. [14] that was obtained in male and female individuals of the house mouse (n = 6 in each sex group). We manually associated the current gene names to the 457 proteins that were detected by mass spectrometry and eliminated those for which there was no correspondence in our scRNAseq dataset. The resulting 370 tear components were plotted against their mRNA level in the total LG epithelium. For proteins that could not be identified individually (e.g., SCGB2B7_SCGB2B20 and SCGB1B20_SCGB1B29), we combined the expression level of the corresponding genes. To identify the tear products most highly expressed by the LG epithelium, we selected those with a normalized expression >15. Differences between males and females were considered significant at the protein level and at the mRNA level if the fold change > 1.5 and adjusted *p*-value < 0.05, based on the original publication [14].

### 2.9. Statistical Analysis

For differential expression analysis, we used Seurat’s default nonparametric Wilcoxon rank sum test, and the *p*-values were adjusted based on the Bonferroni correction using all features in the dataset. Only differences with *p*-adj < 0.05 were considered statistically significant.

## 3. Results

### 3.1. Generation of a Single-Cell Atlas of the Mouse Lacrimal Gland

To generate a comprehensive cell atlas of the LGs, we performed scRNAseq on pooled cells from female (one sample consisting of three animals) and male (two samples, each prepared from three animals) mice using the 10× Genomics platform. Data preprocessing consisted in excluding the doublets and low-quality cells and in reducing the ambient RNA contamination (see Section 2, Appendix A). The integration of the female (7056 cells) and male (9216 cells) datasets resulted in a total of 16,272 cells for cell-type characterization. Unsupervised clustering and uniform manifold approximation and projection (UMAP) analysis revealed 12 main cell clusters within the LG (Figure 1A), which were present in both male and female LGs (Appendix A). The cell clusters were further characterized based on their expression level of the established markers (Figure 1B,C) in combination with their computed marker genes (Appendix A).

Six epithelial clusters were identified based on known markers of the LG epithelium (Figure 1B) [4,5]: *Fxyd3*, *Epcam*, *Sox9*, and *Pax6* (Figure 1C,D). Acinar cells were enriched in *Lpo*, *Prom2*, and *Pigr* [4,27] (Figure 1B,C). The ductal cluster expressing *Wfdc2* was also marked by *Krt14* and *Krt5*, characteristic of basal ductal cells (Figure 1B,C) [5]. A cluster labeled by the luminal ductal markers *Ltf* and *Sftpd* [8,28] was annotated as *Ltf+* epithelial cells, since the presence of a *Lpo+* subset suggested it may also contain acinar cells (Figure 1B,C). Myoepithelial cells (MECs) were identified by the expression of *Krt5/14* and genes related to smooth muscle (Figure 1B,C): *Cnn1*, *Tagln*, and *Acta2* (see Figure 1D for specific ACTA2 immunoreactivity).

Six main clusters of stromal cells were labeled by *Vim* (Figure 1C). They included fibroblasts expressing *Col1a1*, *Apod*, *Lum*, and *Pdgfra* (Figure 1B,C) that are scattered throughout the LG tissue (Figure 1E). Vascular endothelial cells were marked by *Pecam1*, *Flt1*, *Kdr*, and *Ptprb*, while the dual expression of *Acta2* and *Rgs5* indicated the presence of vascular mural cells [29] (Figure 1B,C). The immune compartment was labeled by the pan-leukocyte marker *Ptprc* (Figure 1C), and *Cd52* (Figure 1B)*. Cd52* encodes a glycoprotein expressed by lymphoid cells (*Ms4a4b*, *Trbc2*) and by myeloid lineages (*Cd74*, *H2-Ab1*). The myeloid clusters included *Csf1r*-expressing monocytes and macrophages (Mφ, specifically labeled by *C1qa*), as well as dendritic cells expressing *Cd209a* (Figure 1B,C). Both CSF1R+ Mφ/Mo and CD3+ T cells are commonly found in the normal LG (Figure 1F).

We did not detect any gene coding for hemoglobin subunits, suggesting a successful depletion of erythrocytes from the cell suspension.

### 3.2. Identification of the Stromal Cells Composing the Lacrimal Gland

#### 3.2.1. Vascular Cells

The circulatory system of mammalian tissues consists of venous, arterial, and capillary blood vessels formed by endothelial and mural cells [30].

Mural cells are wrapped around the vascular tube formed by endothelial cells and can either be vascular smooth muscle cells (VSMC) or pericytes (PC). PCs cover capillaries, while VSMCs are associated with larger blood vessels. Consistent with a contractile phenotype, mural cells highly expressed many contraction-related genes (*Acta2*, *Myh11*, *Tagln*, *Tpm1*, *Tpm2*, *Mylk*, and *Cald1*) (Figure 2A). Analysis of the *Acta2*-GFP reporter mouse showed the distinctive morphologies of PCs and VSMCs in the LG, depending on the type of vessel they interact with (Figure 2B). In addition to *Rgs5*, the mural cells in our dataset specifically expressed *Des*, *Notch3*, *Mustn1*, *Gucy1a1*, and *Pdgfrb* consistent with PC identity [29,31,32] (Figure 2C). Together with MECs, they also featured high expression of the matricellular protein *Sparcl1* that regulates angiogenesis and blood vessel integrity [33] (Figure 2C). Although *Myh11* is associated with VSMC identity [29], this cluster of mural cells was not enriched in *Myocd* or *Cnn1* (Figure 2A,C), thus further suggesting it mainly consists of *Acta2+*/*Myh11+* PCs transcriptionally similar to those identified in the kidney [32].

Using the markers previously identified in a cross-tissue single-cell atlas of mouse endothelium [34], we identified three subsets within the vascular endothelial cell (EC) cluster (*Ptprb*, *Pecam1*, *Flt1*, *and Kdr*), namely, the capillary (cap., #1), venous (ven., #2), and arteriolar (art., #3) endothelial cells (Figure 2D and Appendix A). The capillary EC highly expressed *Plvap*, the chemokine *Cxcl12*, and the regulator of cell cycle *Rgcc* (Figure 2E) that was also found in the choriocapillaris that fosters retina [35]. While the arteriolar EC did not express *Pvlap*, they were defined by high levels of *Fbln2* and *Crip1* (Figure 2D) and were specifically enriched in genes controlling endothelial cell identity and vascular integrity (*Cldn5*, *Stmn2*, and *Clu*) [36,37] (Figure 2D). The venous EC were identified by the restricted expression of genes modulating leukocyte adhesion, trafficking, and recruitment (Figure 2E): *Selp* (also known as CD62P) expressed by activated endothelial cells [38], *Ackr1* (DARC1) that binds cytokines [39] and specifically labels venular endothelial cells [40], and *Cxcl10* and *Vcam1* that mediate leukocyte-EC interaction during inflammation [41,42,43]. Finally, the venous EC identity was further supported by the expression of *Vwf*, a marker of postcapillary high endothelial venules [44].

#### 3.2.2. Fibroblasts

Fibroblasts have been identified based on the expression of *Pdgfra*, *Col1a1*, *Lum*, and *Apod* (Figure 2F) [45]. In a recent publication, Buechler and coauthors [46] analyzed 28 datasets and identified several universal subtypes of fibroblasts. We thus adopted the system presented in their study to annotate the four fibroblast subclusters we found in the LG (Figure 2F and Appendix A). Similar to other tissues, the parenchymal fibroblasts (#1, 2) expressed *Penk*, *Crabp1*, *Col15a1*, and *Cxcl14* (Figure 2F), but one large subgroup (#1) was enriched in *Fgf*10+ cells (Figure 2G) that play a pivotal role in LG development, homeostasis, and regeneration [47,48,49]. Adventitial fibroblasts (#3) were specifically enriched in *Ackr3*, *Ly6c1*, and *Dpp4* expression (Figure 2G) and showed high levels of *Pi16*, *Fn1*, *C3*, and *Cd34* (Figure 2F), consistent with a progenitor-like phenotype [46]. These cells located in vascular niches can produce extracellular matrix and can specialize upon stimulus [50,51,52,53]. The *Fgf10^low^* cluster of fibroblasts (#2) had an intermediate transcriptional profile between *Fgf10+* parenchymal and adventitial fibroblasts; they likely represent a transitional stage during fibroblast differentiation. The last cluster #4 was enriched in fibrogenic factors such as *Mfap4*, *Cilp*, *Tnc*, *Pdgfrl*, and *Fxyd6* and was not described as a universal subtype [46]. Therefore, its significance in steady-state and pathological LG remains to be determined.

#### 3.2.3. Lymphoid Cells

Similar to mucosal surfaces, the LG contains multiple cells of lymphoid and myeloid lineages [54] that were largely represented in our cell atlas.

Subclustering of the lymphoid cells identified five subclusters (Figure 3A and Appendix A). The restricted expression of *Gata3*, *Rora*, *Nr4a1/3*, *Areg*, *Csf2*, and *Vps37b* indicated innate lymphoid cells (ILC, #1). In cluster #2, the co-expression of T cell markers (*Cd3g*, *Cd3d*, and *Trac*) combined with higher levels of the retention marker *Cd69* as well as *Vps37b*, *S100a4*, *Ltb*, and *Il7r* was suggestive of resident memory T cells (Trm). A small subset of cells co-expressing *Tcrg-C1* and *Il17a* also indicated the presence of γδT cells, which was confirmed by flow cytometry (Figure 3B). By contrast, the *Cd3g+* cluster #3 showed expression of *Cd8b1*, *Nkg7*, and *Ccl5* revealing cytotoxic T cells (Tc). The expression of *Ly6c2* (involved in memory T homing) suggested an “effector memory-like” phenotype for this cluster enriched in Tc. Lastly, the NK cells (#4) were identified based on the expression of *Nkg7*, *Tyrobp*, *Xcl1*, *Gzma*, and *Klra8*. Proliferating lymphocytes (#5) were defined by the expression of mitotic genes including *Mki67*, *Stmn1*, and *Pclaf*.

#### 3.2.4. Myeloid Cells

The LG contains numerous *Csf1r+* macrophages (Mφ) of different shapes and sizes that closely interact with epithelial cells (Figure 3C). In addition, flow cytometry analysis evidenced a significant number of MHC-II+(encoded by *H2-Ab1*)/ITGAM+/ITGAX+ dendritic cells in the healthy LG (Figure 3D). To gain a better knowledge of the *Cd74+* myeloid subtypes composing the LG, Mφ and monocytes/dendritic cells (Mo/DC) clusters were further divided into finely resolved subpopulations (Figure 3E and Appendix A).

In the Mφ population expressing the canonical markers *Csf1r*, *C1qa*, *Mafb*, and *Adgre1* (Figure 3E), cluster #1 did not have any distinctive features compared to other subgroups but expressed relatively higher levels of heat-shock proteins (Appendix A). This could suggest that these cells are stressed or undergoing differentiation [55]. High levels of *Cd14* and several proinflammatory factors (*Il1b*, *Tnf*, *Nlrp3*, and *Ccl4*) in cluster #2 were suggestive of a polarization into M1 Mφ, which are known for their proinflammatory properties (Figure 3E). Cluster #4 displayed a high expression of *Trem2*, which is associated with the specification of lipid-associated Mφ called foamy cells (or lipid-laden Mφ) by driving a gene expression program related to phagocytosis, lipid catabolism, and energy metabolism [56,57,58]. In addition, this population also expressed high levels of *Lgals3*, *Ctsb*, and *Ctsd* (Figure 3E and Appendix A), similar to the anti-inflammatory TREM2^hi^ foamy Mφ identified in atherosclerotic aorta [59]. In comparison, cluster #5 expressed lower levels of *Trem2* and *Lgals3* but harbored higher levels of *Grn* and also expressed lipid-associated genes (*Pltp* and the phagocytic receptor *Cd36*). Taken together with the specific expression of *Mrc1*, *Pf4*, and *F13a1*, we defined these cells as M2-like resident Mφ.

Population (#10) lacked Mφ markers but expressed *Csf1r*, *Clec4e*, *Plac8*, and high levels of *Lyz2* (Figure 3E) consistent with monocytes. The transcriptional profile of this cluster (*Cx3cr1-*/*Ly6c2+*/*Ccr2+*) suggests they are mostly classical monocytes [60]. DCs in the LG were identified by the expression of *Cd209a*, *Il1b*, and *S100a4*. The LG also contained small populations of Type 1 conventional DCs (cDC1, #8) characterized by *Irf8*, *Naaa*, and *Xcr1*; plasmacytoid DC (pDC, #9), defined by *Plac8*, *Siglech*, and *Ly6d*; and migratory DCs (mig DC, #11) showing the specific expression of *Fscn1*, *Eno3*, and *Cacnb3*. We also identified proliferating Mφ (#3) and DCs (#6) based on the expression of *Birc5*, *Pclaf*, and *Top2a.*

In summary, our data reveal the LG to be composed of a diverse array of immune cells including infiltrating and resident subtypes. The myeloid lineage is preponderant and mainly consists of a heterogenous population of unpolarized and polarized Mφ.

### 3.3. Characterization of LG Epithelial-Cell Clusters

The subclustering analysis of epithelial clusters identified in Figure 1A,B revealed an unexpected complexity for the LG epithelium, with most populations being detected in both male and female mice (Figure 4A,B). The main subtypes were identified using established markers, while new subsets were annotated using their most specific markers (Figure 4C and Appendix A). In accordance with the LG histological structure (Figure 4D,E), *Aqp5* labeled most acinar cells expressing tear products (*Lcn11*, *Scgb2b2*, and *Lpo*), as well as small ducts and intercalated ducts that were characterized by the expression of *Ltf*, *Dmbt1*, *Sftpd*, and a subset of *Krt7*+ cells. Consistent with LG immunostaining (Figure 4F,G), *Krt14* marked larger multilayered ducts (*Wfdc2* and *Krt5/7/19*) and MECs *(Acta2* and *Sox10*).

We then undertook the detailed characterization of the known and novel populations using their respective transcriptional profiles (Appendix A) to evidence their specific features and putative new functions in the LG.

#### 3.3.1. Acinar Populations (#0, 1, 6, 7, 8, 9, 10)

Acinar markers including *Pigr* (that enables the transcytosis of IgA produced by plasma cells), *Bhlha15* (encoding the transcription factor MIST1), and *Prom2* (that regulates plasma membrane microdomains) were mostly detected in clusters #0, 1, and 6 (Figure 5A), thus indicating that they correspond to well-differentiated mature acinar cells [4,27].

The two main mature acinar clusters #0 and #1 had unique features (Appendix A). Indeed, cluster #0 was characterized by a relatively high proportion of mitochondrial genes (median = 8.56%). Although the high expression of mtDNA may be the sign of distressed/dying cells, cluster #0 did not display other features associated with low quality cells (i.e., a low number of genes and UMIs) and showed the highest levels of canonical acinar markers (Figure 5A). Moreover, immunostaining with the mitochondrial markers HSP60 (Figure 5B) or TOM20 (Appendix A) suggests that the mitochondrial content is indeed heterogenous across acinar LG cells. Cells in cluster #1 had the highest RNA content (Appendix A) that could suggest the presence of homotypic doublets; however, this feature is also consistent with their exocrine function, similar to acinar cells in other tissues [61]. Differential expression analysis (Figure 5C, Appendix A) showed that compared to cluster #1, cluster #0 expressed higher levels of genes coding for subunits of the mitochondrial respiratory chain complexes (*mt-Atp6*, *mt-Co2*, *mt-Nd1*, etc.), genes involved in protein synthesis and folding (*Rpn2*, *Eef2*, *Ph4b*, *Qsox1*, etc.), specific tear-related genes (*Hp*, *Lpo*, *Dnase1*, *Pigr*, and *Esp15*), and the lncRNA *Malat1* regulating many cellular functions [62]. Cluster #1 showed many ribosomal proteins (Appendix A) and other factors involved in mRNA processing and polypeptide synthesis (e.g., *Btf3*, *Naca*, *Sec61g*, and *Eif1*) (Figure 5C). We also noted the upregulation of *Gm1553* predicted to code for the tear product lacrein [14], the ferritin subunit *Fth1*, *Tmbs4x* that regulates actin polymerization, and *Tpt1* that stabilizes microtubules (Figure 5C). Therefore, we referred to cluster #1 as “synthesizing acinar cells” actively transcribing components of the cellular machinery for intensive mRNA processing, protein synthesis, and trafficking. As cluster #0 was enriched in mitochondrial genes that are necessary for the production of ATP required for active secretion, we labeled these cells as “secreting acinar cells”. Rather than different cell subtypes, we suggest that these two main acinar clusters correspond to two temporally distinct cellular states; both protein synthesis and secretion are energetically demanding processes that are probably uncoupled to meet the cell capacities for ATP production. Consistent with this, one-micron sections of the LG stained with toluidine blue revealed heterogeneity in acinar morphology, and the appearance of cytoplasmic granules that varied in number, size, and color (Figure 5D).

Cluster #6 had an intermediate profile between clusters #0 and 1, both in terms of metrics (Appendix A) and gene expression (Figure 5A), as shown by the shared expression of cytochrome *Cyp2b10* and of *Edem2*, which is involved in endoplasmic reticulum (ER)-associated degradation. We thus propose that cluster #6 may correspond to a transitional state between the two main phases of acinar function (synthesis and secretion).

Cluster #7 had a low number of genes (Appendix A) and lacked the expression of *Cyp2b10* and *Edem2* (Figure 5A) and thus might be acinar cells not fully differentiated into their mature state. It is also possible that this cluster contained acinar cells of technically lower quality.

Cluster #9 had a similar gene expression profile (Figure 5A) but was found only in male LG (Figure 4A,B). Compared to the male clusters #0 and 1 (Appendix A), it expressed a higher level of some secretoglobins (SCGBs), *Car6*, and exocrine-secreted proteins (*Esp24* and *Esp31*). However, the most significantly upregulated transcript was *Gm42418* that overlaps the locus for Rn45s and thus may represent rRNA contamination in this cluster [63].

Cluster #8 shared most of acinar markers found in other clusters (Figure 5A) but had a unique expression profile (Figure 5E). It contained cells with high expression levels for the transferrin receptor *Tfrc*, transcriptional regulators (*Etv1*, *Malat1*, *Neat1*, *Auts2*, and *Tnrc6a*), and genes involved in mRNA stabilization and splicing (*Tent5a*, *Luc7l2*, *Srsf5*, and *Snrnp70*), thus suggesting that these acinar cells were undergoing transcriptional remodeling. Although high expression of *Malat1* in the scRNAseq data might indicate low quality dying cells, this cluster did not harbor other markers of stress or apoptosis (e.g., heat-shock proteins and high levels of mitochondrial genes). In addition, this *Tfrc*^hi^ cluster expressed higher levels of *Nktr*, which is also found in the kidney and colon epithelium (Human Protein Atlas, www.proteinatlas.org), of the extracellular component *Fgg* (fibrinogen), of the transporter for long-chain fatty acids *Slc27a2*, and of the long noncoding RNA (lncRNAs) *Gm26917* that promotes the survival of satellite cells [64]. Recently, a putative acinar precursor characterized by high expression of *Etv1*, *Neat1*, and *Gm26917* was identified in irradiated parotid glands [65]. Together with a relatively low RNA content (Appendix A), the *Tfrc*^hi^ cluster might correspond to cells with minimal secretory function, which could suggest incomplete differentiation or a functionally quiescent state.

While *Lcn11*, *Lpo*, and *Sval2* were detected at higher levels in the entire acinar compartment compared to ducts and MECs, cluster #10 showed a reduced expression of other acinar transcripts such as members of the ESP family (*Esp15*) of SCGBs (e.g., *Scgb2b2* and *Scgb1b24*), mucin-like protein 2 (*Mucl2*), and the antibacterial *Wfdc12* (Figure 5A). Nonetheless, most of cells in this cluster expressed *Etv1* and the water channel *Aqp5* (Figure 5E,F). They were negative for ductal markers (*Krt7*, *Wfdc2*, *Krt19*, and *Krt14*), and some of them were positive for *Sox10* (Figure 4C). Cluster #10 displayed a particularly high expression for some proteins abundant in tears (*Obp1a*, *Mup4*, and *Bglap3*) (Figure 5F). Strikingly, it was characterized by high levels of *Car6* (carbonic anhydrase 6, CA-VI,) *Dnase1*, and *Kcnn4* that encodes the calcium-activated potassium channel Kca3.1 (Figure 5F). CA-VI immunoreactivity was specifically found in a minority (10%) of acinar cells of the rat LG [66]. Although only ductal cells were immunoreactive for Kca3.1 in the rat LG, *Kcnn4* mRNA was detected in both the ductal and acinar compartments of mouse LG [67,68]. Interestingly, we also found in this cluster a high expression for *Lrrc26* that enables resting K+ efflux in LG epithelial cells by regulating BK channels (large-conductance, voltage, and calcium-activated potassium channel) [69]. Finally, *Car6^hi^* cells were enriched in folate receptor *Folr1* that is expressed at the apical part of polarized epithelial cells [70] (Figure 5F). Altogether, these results suggest that the *Car6^hi^* acinar population has specific functions in the modulation of the protein and ion composition of the tear fluid. We also noticed a small *Car6^hi^* subpopulation that expressed low levels of some of these genes (Figure 5F). This subset of cells was specific to the female LG (Appendix A) and specifically expressed very high levels of a few antimicrobial tear proteins (*Bpifa2*, *Scgb2b27*), *Ggh* that produces folate, amylase *Amy1* secreted by the LG into tears [71], and the immunoregulators *BC037156* and *Smr3a* (Appendix A).

#### 3.3.2. Myoepithelial Cells (MEC)

We previously showed on the whole LG atlas that genes modulating the cellular shape and contraction were exclusively expressed by MECs and mural cells (here, mostly pericytes, PCs) (Figure 2A). The MECs also shared with PCs the expression of the smooth muscle filamin A (*Flna)* and of *Csrp1*, which is a transcription factor associated with the smooth muscle lineage [72] (Figure 6A). Contrary to PCs, the MECs exhibited expression of *Cnn1* (Figure 2A) and the calmodulin regulator *Pcp4* (Figure 6A). They were also characterized by the expression of *Igfbp2* and *Igfbp5* that regulate the response of smooth muscle cells to insulin-like growth factor 1 [73] (Figure 6A). Together with the main ductal cluster, the MECs expressed genes important for tissue development and homeostasis (Figure 6A): *Trp63*, which encodes the N-terminally truncated isoform of p63 (ΔNp63) known to preserve the self-renewal capacity of stem cells in glandular structures [74], and *Net1*, which is necessary for proper ductal branching and the MEC function in the mammary gland [75].

The MECs also expressed proteins of tight junctions (*Nectin1*), components of the basement membrane and proteoglycans (*Lamb3*, *Col4a1*, *Col4a2*, *Bgn*, and *Dcn*) and cell–ECM interaction molecules (*Sparc*, *Sparcl1*, *Ccn1*, *Ccn2*, *Mia*, *Lgals1*, *Thbs1*, and *Sema5a*) that regulate multiple processes including proliferation, differentiation, migration, angiogenesis, etc. (Figure 6B).

Strikingly, we found that the MECs featured many other communication molecules and secreted factors (Figure 6C). They specifically expressed cholecystokinin (*Cck*) encoding a hormone that regulates pancreatic secretion [76], neuregulins (*Nrg1* and *Nrg2*) that modulate cell growth and differentiation, and neurturin (*Nrtn*) that promotes the survival of neurons [77]. According to our data, MECs likely secrete factors regulating cell proliferation (*Gas1* and *Gas6*) and pleiotropic molecules (*Pdgfa*, *Mdk*, and *Sfrp1*), including *Kitl* that promotes stem cell maintenance. They also expressed high levels of *Pgf* involved in endothelial cell growth and angiogenesis, as well as the lymphokine *Mif* also found in all ductal cells. While MECs possess long processes that surround the acini, they also physically interact with cells composing the stroma, including neuronal (Figure 6D) and vascular (Figure 6E) cells.

Therefore, our scRNAseq data introduce a large panel of genes supporting a unique role for MECs in ECM organization and cell–cell communication, beyond their contractile function and the maintenance of acini integrity.

#### 3.3.3. Ductal Populations (#3, 4, 5)

Three clusters (#3, 4, and 5) contained cells expressing the ductal marker *Krt7* (Figure 7A).

Clusters #3 and #5 expressed the cytokeratins *Krt5/14/17* marking the basal ductal cells and MECs in the adult LG (Figure 7A) [5,78]. Both clusters also shared with the MECs the expression of the calcium sensor *S100a6*. Nonetheless, the ductal clusters #3 and 5 specifically expressed *Ly6d* that labels luminal progenitors in the prostate, *Ly6a* (also known as Sca-1) that is found in many stem cells but also in mammary ductal cells and tubular cells in the kidney [79,80], and the hormone subunit *Gpha2* found in other glands (pancreas and pituitary gland) and in the corneal limbal stem cells that renew corneal epithelium [81,82] (Figure 7A). These features support our previous cell-lineage tracing studies showing the existence of slow-cycling stem/progenitor cells in the basal ductal compartment of multilayered ducts [4,5].

Cluster #5 had a slightly higher proportion of *Krt19+* and *Wfdc2+* cells compared to cluster #3, suggesting that it may be enriched in luminal ductal cells (Figure 4B). Cluster #3 expressed higher levels of the intestinal stem cell marker *Smoc2*, the transcription factor *Klf6*, and secreted signaling factors (e.g., *Ccn1*, the chemokine *Cxcl1*, and the stem cell factor *Kitl*) (Figure 7A). The differential expression analysis between cluster #3 and #5 evidenced many genes related to inflammation (*Cxcl1*, *Nkbiz*, *Nfkbia*, etc.) and stress response (*Atf3*, *Egr1*, *Gadd45b*, etc.) upregulated in cluster #3 (Figure 7B, Appendix A). Since some of the differentially expressed genes were described as induced by the tissue dissociation process, we used this previously established list of dissociation-induced genes [83] and found that cluster #3 did have the highest score for this signature (Figure 7C, dotted circle). Therefore, it is not excluded that some of the differences found between these two ductal clusters may also reflect different levels of stress following enzymatic digestion. However, it was recently shown in the single-cell atlas of the developing salivary gland that several of these transcription factors (*Junb*, *Nfkbia*, *Klf6*, *Atf3*, and *Fos*) are involved in duct development and belong to basal ductal identity [84]. Therefore, cluster #3 may be enriched in basal ductal cells and play a major role in paracrine signaling in the LG.

Cluster #4 had low to no expression of *Wfdc2* and *Krt5/14/17* (Figure 7A). However, this cluster expressed high levels of *Aqp5*, the antimicrobial tear product *Wfdc18*, and *Krt18* that is highly expressed in ducts [85] but also found in a subset of acinar cells in rat and human LG [86] (Figure 7D). Cluster #4 also harbored a very unique profile including the specific expression of *Ltf*, *Dmbt1*, and its binding partner *Sftpd* that is expressed by LG ductal cells [4,28] (Figure 7D). Lactotransferrin (LTF) is an iron-binding protein that has antimicrobial properties and is found in ducts and discrete acinar subpopulation in the mouse LG [8]. Taken together, cluster #4 mainly contains luminal cells, likely of smaller ducts and intercalated ducts, and possibly few cells of the acinar lineage. Interestingly, we found that this *Ltf+* cluster was also marked with the ferritin subunits (*Fth1* and *Ftl1)* that mediate intracellular iron storage, thus suggesting that these cells accumulate iron (Figure 7D). Furthermore, this cluster expressed the highest levels of *Nupr1* that can repress an iron-dependent cell death called ferroptosis [87]. Notably, several specific markers associated with immune function (*Pglyrp1*, *Ccl9*, and *Cxcl17*) and the modulation of fluids (*Fcgbp* and *F5*) have been identified in this *Ltf+* cluster (Figure 7D), therefore supporting unique roles in communication with the immune system and the modification of the primary tear fluid.

### 3.4. scRNAseq Profiles Several Kit+ Progenitor Subsets in the Lacrimal Gland

We previously showed that c-KIT+^dim^/EpCAM+/Sca1-/CD34-/PTPRC- LG cells after transplantation were able to generate both ductal and acinar cells in vivo [7]. *Kit*-expressing cells (Figure 8A) in clusters #3 and 5 accounted for only 4–5% of the ductal cells. They probably correspond to the cKIT+/KRT14+ label-retaining cells that we previously identified in the basal ductal compartment of the mouse LG [5]. The *Car6^hi^* acinar cluster (#10) and the *Ltf+* cluster (#4) contained the highest proportion (21% and 24%, respectively) of *Kit*-expressing cells (Figure 8A), suggestive of a distinct KRT14-/KIT+ progenitor group, similar to mouse salivary glands [88].

Previous studies on human LG have suggested that c-Kit+ epithelial LG progenitors can be identified by high ALDH1 activity [89] that detoxifies the aldehydes derived from lipid peroxidation. Interestingly, we found a subpopulation positive for *Aldh1a1* within the *Ltf+* cluster (Figure 8B). Over 50% of the *Aldh1a1+* cells in the *Ltf+* clusters also expressed *Kit* (Figure 8C). By comparing the *Aldh1a1*+ and *Aldh1a1*- cells within the *Ltf+* cluster (Appendix A), we found that the *Aldh1a1*+ cells had significantly higher expression levels for genes that protect against oxidative damage including *Gsta4*, *Gpx2*, and *Prdx3* (Figure 8C, Appendix A). Interestingly, the *Aldh1a1*+ cells also had increased expression for carbonic anhydrases *Car12* and *Car2* that regulate acid–base homeostasis (Figure 8C). In our dataset, *Car2* was mostly detected in the *Ltf+* cluster (Figure 8D). *Car2* encodes CA-II, which was found in a few acinar cells and in cells located in the initial segment of the intercalated duct in the mouse LG [90]. In the pancreas, *Aldh1a1* and *Car2* mark pancreatic progenitor cells, called centroacinar and terminal duct cells (CA/TD) [91,92], which are located at the junction between acini and the duct. In the mouse LG, we previously found a subset of putative epithelial progenitors expressing the transcription factor *Runx1* that we described as centroacinar-like cells [93]. *Runx1* regulates LG morphogenesis [93], and its expression was detected in 73% of the *Aldh1a1+* cells (Figure 8E). In addition, the *Aldh1a1*+ cells expressed many other factors suggestive of progenitor cells (Figure 8E). They were indeed characterized by the expression of *Cd14* that together with *Kit* and *Aldh1a3* is expressed in mammary epithelial progenitors [94]. *Aldh1a1*+ cells also had significantly higher levels of the transcription factors involved in developmental programs such as *Barx2* that are necessary for the chemotactic response of epithelial cells to FGF10 during LG morphogenesis [47], key regulators of duct maturation in other glands (*Tfcp2l1* [95], *Nfia* [96], and *Nfib* [97]), and *Sox9* that is required for LG branching [98]. They expressed several tetraspanins (Appendix A), which are involved in the regulation of stemness in other cell types [99], including *Tspan8* that is upregulated by SOX9 in pancreatic ductal adenocarcinoma in response to EGF stimulation [100] (Figure 8E). Finally, in addition to pathways related to redox balance and developmental processes, *Aldh1a1+* cells were significantly enriched in the genes involved in lipid metabolism, as shown by the top-listed pathways entitled “biological oxidations”, “fatty acid derivative biosynthetic process”, and “fatty acid oxidation”, for instance (Appendix A), thus supporting a specific metabolic profile for this population. We also noted significant pathways related to ion transport such as the “transport of small molecules” and the “regulation of potassium ion transmembrane transporter activity”. The *Aldh1a1+* cells indeed displayed higher levels of several ion transporters/channels (*Slc12a2*, *Atp1b3*, *Slc5a8*, and *Clic6*) (Figure 8E)*. Slc12a2* codes for NKCC1 that is widely expressed in luminal ductal cells of the lacrimal and salivary glands [4,67]. Intriguingly, *Clic6* was detected only in this subpopulation of the LG epithelium, suggesting it may serve as a specific marker (Figure 8F). The *Aldh1a1*+ cells of the *Ltf+* cluster may thus mediate ion transport, another feature shared with pancreatic CA/TD that plays an important role in regulating the ionic composition of the ductal lumen [101].

SOX10 is a pivotal transcription factor for the specification of KIT+ progenitors and is required for LG branching and morphogenesis upon FGF10 stimulation [6,98]. It is established that *Sox10* marks the acinar and MEC lineages of the LG [4,5]. *Sox10* was indeed abundantly expressed by MECs (Figure 8G), but double-positive *Kit+*/*Sox10+* cells were almost exclusively found in the *Car6^hi^* and the *Ltf+* clusters (Figure 8H). Short-term administration of tamoxifen (TM) in the *Sox10*-iCre:TOM adult mouse labeled MECs and acinar cells and, unexpectedly, some ductal and intercalated duct cells (identified as E-CAD+/PECAM-1-) (Figure 8I). Consistent with previous studies, lineage tracing over one week following the TM injection labeled MECs and acini especially at the distal part of the LG (Appendix A). Clusters of TOM+ acinar-like cells were also observed at the junction between the acini and intercalated ducts (Appendix A). In addition, we observed TOM+ clones in intercalated ducts (id) (Appendix A) and multilayered excretory ducts (ed) (Appendix A). Altogether, this shows that MECs, acinar cells, and a subset of ductal and intercalated duct cells are derived from adult SOX10+ populations. SOX10 immunostaining in *Sox10*-iCre:TOM adult mouse LG one week after TM injection revealed that large TOM+ acinar cells do not harbor SOX10 nuclear staining, thus suggesting they have shut down SOX10 expression at the time of sacrifice (Figure 8K and Appendix A). By contrast, MECs and most cells located in the intercalated ducts and at the junction with acini were stained by the SOX10-antibody (Figure 8J,K and Appendix A). Small SOX10+ acinar cells were also observed in the central region of the acini, eventually surrounded by clusters of large SOX10- acinar cells (Figure 8K and Appendix A). We thus propose that mature SOX10- acinar cells originate from SOX10+ cells located in secretory units (acini and intercalated ducts) and that larger ducts contain a subset of SOX10+ cells that contribute to their development and maintenance, similar to embryonic LG development [6,98]. Therefore, the *Ltf+* cluster likely contains *Kit+/Sox10+* progenitors capable of differentiation towards acinar cells (at the terminal end of intercalated ducts) and some luminal cells, while the *Kit+/Sox10+* cells of the *Car6^hi^* cluster probably correspond to immature acinar cells similar to those observed in the vicinity of the central lumen (Figure 8K and Appendix A).

In summary, we propose that the *Ltf+* cluster contains two distinct putative progenitor populations: a luminal *Aldh1a1+* population that shares similar properties with pancreatic CA/TD cells and a subset of *Kit+/Sox10+* epithelial progenitors that are mostly found in intercalated ducts. On the other hand, the *Car6^hi^* cluster most likely contains committed SOX10+ acinar progenitors that may represent a reservoir for tissue repair.

### 3.5. Gene signatures of Discrete Adult Populations Correlate with Developing Epithelial Populations of the Postnatal Lacrimal Gland

If indeed the *Tfrc^hi^* are immature acinar cells, and the *Car6^hi^* and *Ltf+* clusters contain committed progenitor cells that can contribute to LG maintenance, it is reasonable to expect that some of the populations undergoing differentiation in the developing LG would exhibit similar features. To test this hypothesis, we reanalyzed the gene expression profile of the LG epithelium during postnatal development using previously published scRNAseq data [4] from mouse LG collected at postnatal day 4 (P4) (Figure 9A).

The annotation of all cell types (Appendix A) was conducted using the markers list generated by unsupervised clustering (Appendix A) and inspection of a signature for mitotic genes (Appendix A). By investigating the key features discovered in the stromal clusters of the adult LG, we found that, in spite of the relatively low number of cells, this dataset of early postnatal LG faithfully recapitulated the fibroblastic (Appendix A), immune (Appendix A), and vascular (Appendix A) populations observed in the adult LG. Clusters of the developing epithelium (Figure 9B) were identified by the expression of *Krt18*, *Epcam*, and *Krt8* (Figure 9C). The cluster of *Krt8*+/*Krt18*+ cells that did not harbor clear ductal nor acinar identity was annotated as “undifferentiated” epithelial cells (Figure 9B,C). Proliferating epithelial cells were evidenced by a higher score for the G2/M gene signature (Appendix A). MECs formed a clear separate cluster of cells expressing *Acta2*, *Krt14*, and *Igfbp2* (Figure 9C). The ductal compartment was labeled by *Krt14*, *Krt7*, and *Aqp5*, while the acinar lineage (differentiating acinar and *Dcpp+* epithelial cells) was identified as *Krt7-/Aqp5+* (Figure 9C).

Differentiating acinar cells expressed tear products: *Pip*, *Mucl2*, *Scgb2b2*, and *Scgb2a2*, for instance (Figure 9D). Interestingly, cells expressing high levels of both *Tfrc* and *Fgg* were detected in this cluster (Figure 9E), similar to the *Tfrc*^hi^ acinar cells of the adult LG. *Etv1* was not a major feature of these differentiating acinar cells; however, it was detected in another small cluster marked by a very strong expression of the *Dcpp* family of genes that are involved in acinar differentiation in the salivary glands [102] (Figure 9F). These *Dcpp+* epithelial cells were positive for *Scgb2b27*, *Folr1*, *Wfdc18*, *Bpifa2*, and *Car6* and expressed the potassium channel *Kcne1* (Figure 9F), thus intriguingly resembling the adult *Car6^hi^* acinar cluster. In addition, they featured the high expression of *Pglyrp1* and *Dmbt1* that we previously found specifically in the adult *Ltf+* cluster (Figure 9F).

A small subset of cells in the ductal compartment was labeled by *Ltf*, *Sftpd*, *Nupr1*, and *Clic6* (Figure 9G). Interestingly, the ductal compartment also expressed the progenitor markers *Kit*, *Barx2*, and *Runx1* and markers for putative CA/TD cells (*Car2* and *Aldh1a1*) that we previously identified within the adult *Ltf+* epithelial cluster (Figure 9H).

To summarize, the immature epithelial populations observed at P4 exhibit similarities with subpopulations of the adult LG, particularly the *Car6*^hi^ and *Ltf*+ putative progenitors. The detection of their key marker genes during the early postnatal development of the LG strengthens our hypothesis that these discrete cell subsets may contribute to the expansion and maintenance of the adult LG epithelium.

### 3.6. Acinar Cells Are Not the Only Source of Tear Proteins and Sex-Biased Factors

While composition of the tear film has previously been studied in rodents and humans, little is known about the exact cellular origin of tear proteins and the role of LG cells in the sex differences found in tears.

To address this, we used the previously published data about the tear proteome of the house mouse [14] and selected all proteins for which we detected the mRNA in our dataset (Appendix A). Protein signal intensities obtained by mass spectrometry were plotted against their mRNA expression level in the LG epithelium to identify tear proteins that are most likely expressed by LG cells (Appendix A). We retained 23 tear products with a relatively high expression level in the LG epithelium (average expression value > 15, in red on Appendix A).

The abundantly expressed proteins included ACTB, APOE, and TPT1. Their transcripts were found at higher levels in MECs and ductal clusters (Appendix A). These proteins are broadly expressed in tissues and many cell types. Therefore, they are unlikely to be specific to the LG secretome. Moreover, intracellular ACTB likely originates from cells damaged during the tear collection procedure. Of note, the proteins mainly expressed by ductal cells and MECs (WFDC2, WFDC18, and LNC2) (Appendix A) that are found in saliva [103,104] were barely or not detected in tears. This might be explained either by the technical limitations or by the fact that they might not be released into the tear fluid.

Three out of the twenty-three tear components identified were maximally expressed by other clusters than the mature acinar clusters: *Ltf*, *Car6*, and *Esp18* (Appendix A). Ductal lactoferrin (LTF), CA-VI, and ESP18 were significantly enriched in male tears (Figure 10A) [14]. While we detected similar *Ltf* expression in both males and females, the *Ltf+* epithelial cluster was smaller in female LGs (Figure 4B), and *Ltf*-expressing cells were also found in smaller proportions within the cluster of females (53% compared to 90% in male *Ltf+* cluster) (Figure 10B). Therefore, a reduced population of *Ltf+* ductal cells in female LG might explain the difference observed between male and female tears. A similar assumption is made regarding male-enriched CA-VI, as a smaller proportion of female cells have a high *Car6* expression level within the cluster #10 (Figure 10A,B). In our dataset, the *Ltf+* and *Car6^hi^* clusters also expressed slightly higher levels of *Esp18* compared to other clusters, but the differences we observed at the mRNA level between males and females in the *Car6^hi^* cluster are unlikely to be the cause of a higher abundance of ESP18 in male tears (Figure 10A). In addition, *Esp18* was found at slightly higher levels in other female acinar cells compared to males (Figure 10B).

In accordance with previous assumptions, the highest expression levels for most other tear components were found in acinar cells (Appendix A). The lack of acinar specificity for the expression of these transcripts is likely caused by the cross contamination originating from acinar cells (see Section 2, Appendix A). Acinar clusters #0, 1, 6, and 7 displayed similar expression patterns within each sex group (Appendix A). Therefore, we have grouped them together into a single “acinar compartment” (Figure 10C). Cluster #9 was kept apart from the main acinar compartment as it was exclusively found in male LG and had no specific secretome (Appendix A). Based on their distinct transcriptomic profiles, the *Tfrc*^hi^ (#8) and *Car6*^hi^ (#10) clusters were not included into the main acinar population. However, we did not notice any specific features related to tear secretion in the *Tfrc*^hi^ cluster, apart from a lower expression levels of tear components (except for male *Scgb1b29*) compared to the mature acinar clusters (Appendix A).

Similar to the tear proteome data (Figure 10A), *Esp6*, *Esp15*, *Scgb1b2*, *Scgb2b2*, and *Lcn11* were expressed at similar levels between males and females in the main acinar compartment (Figure 10C). While Stopkova and co-authors [14] found that *Obp1a* and *Pip* were significantly enriched in female tears, their mRNA levels did not differ between male and female acinar cells, thus suggesting the effect of sex on their regulation might be post-transcriptional. In accordance with the protein levels, we found that *Obp1b* was highly specific to females and that *Sval2* was significantly enriched in female acinar cells. While SCGB2B20 is female-biased in tears, its gene was significantly upregulated in male acinar cells compared to females. Stopkova and colleagues reported a similar finding using bulk RNAseq of the LG [14]. Consistent with the tear proteome data, we found many male-biased transcripts: *Mup4*, *Scgb1b3*, *Scgb1b20/29*, and *Scgb2b7/20.* Although the sex differences in the tear levels of ESP16 did not reach statistical significance, we observed that the *Esp16* transcript was exclusively expressed by male acinar cells. *Gm1553* encoding lacrein (LCRN) and *Scgb2b24* were significantly enriched in the male acinar compartment, but no differences were seen in tears.

To conclude, some tear components originate from ductal cells. The enrichment of LTF in male tears is likely due to a larger number of *Ltf+* ductal cells in male LG. Nonetheless, acinar cells are the major contributor for tear proteins and are responsible for additional sex dimorphism regarding tear composition.

## 4. Discussion

In this study, we generated the first comprehensive single-cell atlas of the adult mouse lacrimal gland, described the putative progenitor populations, and investigated the cellular origin of sex differences in tear composition.

Our analysis revealed the complexity of the stromal landscape of the LG. The extensive data obtained regarding the vascular, fibroblastic, and immune subtypes that constitute the LG will be valuable for future studies exploring the common features and unique characteristics of these cells across different tissues. Our study not only allows for the identification of distinct stromal clusters within the LG, but it also provides an opportunity for future studies to compare the expression of cluster-specific markers under both normal and pathological conditions. Consistent with the previous notion that the LG is a part of the mucosal immune barrier system [54,105,106], our study revealed a substantial presence of resident immune cells from both the lymphoid and myeloid lineages in the gland. We recently showed that the epithelial LG cells are sentinel cells that can sense danger signals/tissue damage and recruit immune cells by activating the inflammasome pathway [107]. Therefore, it is not surprising to find a considerable number of innate immune cells and resident regulatory T cells in the LG in homeostatic conditions, as they can rapidly initiate an inflammatory response upon sensing alarm signals.

Aside from their interactions with the immune system, LG epithelial cells directly contribute to the defense of the LG and the ocular surface by secreting many factors that have antimicrobial and/or immunoregulatory properties.

### 4.1. Ltf+ Ductal Cells Contribute to the Defense of the Ocular Surface

Indeed, our study shows that ductal cells, for instance, contribute to the barrier function of the LG. In the past, several studies have demonstrated that ductal cells significantly participate in the release of water and ions into the tear fluid [108]. The presence of numerous secretory granules in the cytoplasm of ductal cells, as seen by electron microscopy, suggests that they may also secrete proteins [109]. In our single-cell atlas of the LG, we found a cluster of luminal/intercalated duct cells specifically expressing *Ltf.* LTF is abundant in tears and able to bind free irons, liposaccharides, and markers of inflammation, thus preventing microbial growth and immune activation [110]. The *Ltf+* cluster was characterized by the expression of *Sftpd*, *Dmbt1*, and *Pglyrp1.* These antimicrobial factors also produced by salivary glands are known to bind bacterial peptidoglycans, indicating a potential role in the recognition and response to bacterial infections [111,112]. *Dmbt1* codes for glycoproteins that can be secreted and are involved in various biological processes including mucosal innate immunity. DMBT1 is present in human tears [113] and has been shown to interact with IgA (produced by plasma cells of the LG), LTF, and SFTPD, thereby enhancing phagocytosis by cells of the innate immune system [114]. Unlike mice, where SFTPD was found in ductal cells [28], human SFTPD is produced by LG acinar cells and detected in tears [115]. Altogether, our findings suggest that mouse luminal ductal cells expressing *Ltf* not only modify the ionic composition of the primary tear fluid but also produce factors that play a role in the host defense. Additionally, these cells produce chemokines, such as CCL9 and CXCL17, that are responsible for attracting innate immune cells to the site of infection or inflammation.

### 4.2. Ductal Cells and Myoepithelial Cells Mostly Have a Local Secretory Function

Furthermore, our analysis revealed that other ductal clusters, which contain both basal ductal cells and some luminal cells of multilayered ducts, exhibited high expression levels of two members of the Whey-acidic-protein (WAP) four-disulfide core domain protein family: *Wfdc2* and *Wfdc18*. These proteins are predicted to possess both inflammatory and antimicrobial properties and have also been detected in human saliva [103]. WFDC2 was not found in tear proteome data [14,16], while WFDC18 was detected in only two males [14], suggesting they are not major components of the tear fluid. To our knowledge, their potential antibacterial function has not been experimentally confirmed [116]. However, *Wfdc18* was shown to promote apoptosis in mammary epithelial cells [117], and *Wfdc2* could be involved in innate immunity and adenocarcinomas [118]. Similarly, the lipocalin LCN2, which is highly expressed by MECs and detected in some ductal cells in our dataset, was not found in the tear proteome data [14]. Due to its capacity to bind hydrophobic ligands, LCN2 has a wide array of functions from antimicrobial properties (through the sequestration of bacterial siderophores [119]) to the regulation of energy metabolism, glucose, and lipid homeostasis [120]. Therefore, these proteins secreted by MECs or ductal cells are either in too small amounts to be detected in tears or may not be released into the tear fluid and rather have a local effect in the LG that may go beyond their predicted antimicrobial activity.

In support of a major local effect for the secretome of MECs, we found that MECs shared with vascular cells or fibroblasts the expression of genes involved in the composition or the organization of the extracellular matrix, as well as previously described factors involved in cell–ECM and cell–cell interactions [121]. For instance, Thrombospondin-1 (*Thbs1*, encoding TSP-1) was specifically expressed by MECs at the transcriptional level. TSP-1 is found in MECs and ducts by immunostaining [7,122], and its deficiency in mice leads to MEC dysfunction resulting in LG inflammation and dry eye [123,124]. Most importantly, MECs secrete pleiotropic signals that regulate multiple cellular processes critical for stem cell maintenance, LG development, and homeostasis. Although some of them were also expressed by ductal or stromal subtypes, MECs seem to be the unique source of paracrine signals such as cholecystokinin and neurotrophic factors (neuregulins and neurturin). By surrounding acini, MECs are in close proximity with intercalated ducts and at the interface between the secretory units and the stroma containing blood vessels, neuronal cells, fibroblasts, etc. Due to their unique transcriptional profile, it is likely that MECs function as niche cells that can signal to any other cell type, thereby regulating LG homeostasis and preserving acinar integrity to ensure proper tear secretion.

### 4.3. Acinar Cells Are the Main Producers of Tear Proteins and Sex-Biased Signals

Comparison of our transcriptomic data with previously published proteomics data confirmed that acinar cells of the LG are the main source for the most abundant tear components, with the exception of LTF. In accordance with previous proteomic studies, we also found that acinar cells are responsible for the secretion of female-biased OBP1b and SVAL2 and of many male-biased tear products (MUP4, SCGB1B20/29, SCGB2B7/20, and SCGB1B3). Discrepancies regarding other tear components might arise from post-transcriptional mechanisms, the eventual contribution of other organs, and strain-related specificities. SVAL2 and the secretoglobins are thought to participate in the barrier function of the epithelium and to modulate immune responses [125,126]. OBP1b and MUP4 are part of the lipocalin family of proteins that can bind small hydrophobic molecules [127]; thus, OBPs and MUPs are involved in chemical communication by transporting volatile pheromones [14,128,129,130]. MUP expression levels depend on multiple factors including sex, social status, housing conditions, and estrus cycle [130,131,132]. Therefore, these parameters likely affect the individual mouse repertoire of lipocalins across different studies, and an in-depth investigation of sexually dimorphic genes would require taking all these factors into account, including variations related to age and the circadian clock [133].

At the transcriptional level, our detailed analysis of the LG epithelium identified two main populations of mature acinar cells annotated as “secreting” and “synthesizing” acinar cells. Secreting acinar cells expressed high levels of *Lpo*, *Hp*, *Prom2*, *Pigr*, and mitochondrial transcripts, while synthesizing acinar cells had a high RNA content and were enriched in ribosomal proteins and lacrein (*Gm1553*). Considering that the synthesis of tear products and the secretory process (including organelle reorganization and membrane recycling) both require ATP and share some of the cellular machineries, we propose that LG acinar cells alternate protein synthesis and secretion, similar to the parotid gland [134]. Furthermore, the desynchronization of acini would enable the continuous production of the tear fluid necessary for the protection of the ocular surface. Consistent with this, we found some heterogeneity in the number of granules and mitochondria across LG acinar cells. Similar to our observations, mitochondria positioning and dynamics are modulated by stimulated exocytosis in secretory acinar cells of salivary glands [135]. Interestingly, “secreting” acinar cells expressed lower levels of ferritin that enables iron storage. By contrast, *Prom2* was enriched in these cells and, although its function is largely unknown, it was shown that PROM2 promotes the formation of ferritin-containing multivesicular bodies and exosomes for iron export in other tissues [136]. Therefore, cellular iron content may be differentially regulated between the two functional phases of acinar cells. Iron is a key factor for many cellular processes such as the tricarboxylic acid cycle, lipid metabolism, and gene regulation; however, it is toxic in excess, and its cellular concentration is thus tightly regulated [137]. High expression of ferritin subunits (*Fth1* and *Ftl1*) and *Nupr1* in the epithelial *Ltf+* cluster suggested iron accumulation in these cells. In humans, lower serum levels of iron are associated with a decreased quality and quantity of tears in term newborns [138]. Further research is needed to clarify whether iron plays a direct role in the secretory process or is solely associated with the production of iron-binding tear products such as haptoglobin (HP) by acinar cells and LTF by luminal cells.

Cellular iron uptake is mediated by the transferrin receptor (*Tfrc*), whose expression was detected in most acinar clusters but was found at higher level in a small group of acinar cells also characterized by high expression levels of *Etv1* and *Fgg*. The ETV family comprises FGF-responsive transcription factors that are involved in the proper formation of the lens and the LG during mouse embryonic development [139,140]. *Etv1* also correlated with acinar differentiation in the developing mouse submandibular gland [84]. We also found co-expression of *Tfrc/Fgg* in the cluster of differentiating acinar cells of the LG at P4. We thus propose that this *Tfrc^hi^* cluster may represent a pool of immature acinar cells either undergoing transcriptional remodeling towards a fully mature secretory phenotype or awaiting activation.

### 4.4. Epithelial Progenitors Exist in the Adult Lacrimal Gland

Previous studies have shown that the LG has stem/progenitor cells capable of LG repair after tissue damage [7,141]. In our cell atlas, we identified *Kit+/Aldh1a1+/Car2* cells similar to pancreatic centroacinar/terminal ductal cells (CA/TD) within the *Ltf+* cluster enriched in intercalated duct cells. This population had a unique metabolic profile and expressed progenitor markers, as well as several ion channels, including their newly identified specific marker *Clic6*. Interestingly, *Clic6* was also expressed by developing ductal LG cells at P4 and by *Kit*+/*Gfra3+* intercalated ducts of the submandibular gland that share key features with the P1 salivary *Krt19+* ducts [84], thus suggesting it may serve as a universal marker for adult ductal progenitors in these glands. *Clic6* was recently discovered as a member of the chloride intracellular channel (CLIC) gene family but did not display chloride channel activity in vitro [142]. Although its precise function remains unknown, it was shown that this metamorphic channel undergoes oligomerization in oxidative conditions and is stabilized by membrane lipids [143]. It is thus possible that the tuning of the redox status of *Aldh1a1+* cells might participate in the modulation of CLIC6 activity, in addition to a potential role in the maintenance of stemness and differentiation processes [144].

Importantly, we also identified epithelial *Sox10+* subsets of distinct lineages. SOX10 is necessary for the establishment of secretory units (including MEC, acini, and intercalated ducts) in several exocrine glands and also mediates epithelial cell plasticity [6]. Athwal and co-authors showed that *Sox10* induction in distal KIT+ epithelial progenitors requires FGF10 stimulation [6]. Using duct cell lines, they also demonstrated in vitro that the effect of SOX10 depends on the growth factors present, promoting either epithelial differentiation or a progenitor-like state maintaining KIT expression. Our study showed for the first time that the adult LG contains SOX+ cells in intercalated ducts and the ductal compartment that likely correspond to *Kit+/Sox10*+ cells of the *Ltf+* epithelial cluster. We also described the existence of small SOX10+ acinar cells close to the central lumen, which might relate to *Kit+/Sox10*+ cells of the *Car6*^hi^ acinar cluster. Since the *Car6*^hi^
*and Ltf+* clusters uniquely share some common features (*Krt18*, *Wfdc18*, *Folr1*, and *Lrrc26*) and that P4 LG’s *Car6+* cells express markers specific to the adult *Ltf*+ cluster (*Dmbt1* and *Pglyrp1*), it is plausible that the *Car6*^hi^
*and Ltf+* clusters are linked and may correspond to different differentiation stages associated with distinct locations in the gland (such as from the proximal to the terminal part of the intercalated duct). It is also possible that they originate from a common *Kit+/Aldh1a1+* progenitor. Further studies are needed to confirm these hypotheses or to determine whether they correspond to distinct pools of reservoir cells with different cell fates.

In our dataset, *Car6*^hi^ acinar cells were characterized by high *Dnase1* and K^+^ channel-related genes’ (*Kcnn4* and *Lrrc26*) expression levels. Cells immunoreactive for CA-VI represent only 10% of rat LG acinar cells. However, to our knowledge, nothing is known about this population. Nonetheless, *Car6*^hi^ cells seem relevant for LG pathology, since autoantibodies for CA-VI are early biomarkers for the diagnosis of Sjogren’s Syndrome [145], and using spatial transcriptomics on mouse LG, we found a rare *Car6*^hi^ population that was particularly affected by Sjogren’s Syndrome-like disease in *NOD.H2^b^* mice [85].

## 5. Conclusions

To summarize, our study provides the first comprehensive atlas of the adult mouse lacrimal gland, as well as the molecular foundation for the comparison of male and female LGs. Altogether, these findings offer new avenues for the study of LG postnatal development, the maintenance of LG cellular homeostasis, and the identification of the populations and mechanisms that control adult LG regeneration. Future comparisons of our data with pathological models will help to understand LG dysfunction and to identify new therapeutical strategies to manage aqueous deficiency dry eye.

## 6. Limitations of the Study

Although we supported our statements using relevant previous studies and aimed to confirm key observations at the transcriptomic level using immunostaining and lineage tracing experiments, some of our findings and hypotheses will require future studies to be validated at the protein level or with functional assays.

In addition, we designed our study so that each sample would combine the LGs from several mice sacrificed at the same time of the day to better represent the mouse LG in general and evaluate the sex differences. Consequently, our data do not offer the possibility of assessing the potential individual variability or the transcriptomic changes that may occur under the influence of the circadian rhythm, estrous cycle, or housing conditions.

While scRNAseq is a powerful tool to interrogate the transcriptome of each individual cell, it suffers some obvious technical limitations related to the tissue dissociation process and ambient RNA cross contamination. For instance, our dataset lacked neuronal populations and plasma cells, and epithelial populations were underrepresented compared to the composition of intact LG. Therefore, future studies will benefit from very recently developed spatial multi-omics techniques, analyzing tissue sections that preserve all cells together with their morphology and interactions.

## Figures and Tables

**Figure 1 cells-12-01435-f001:**
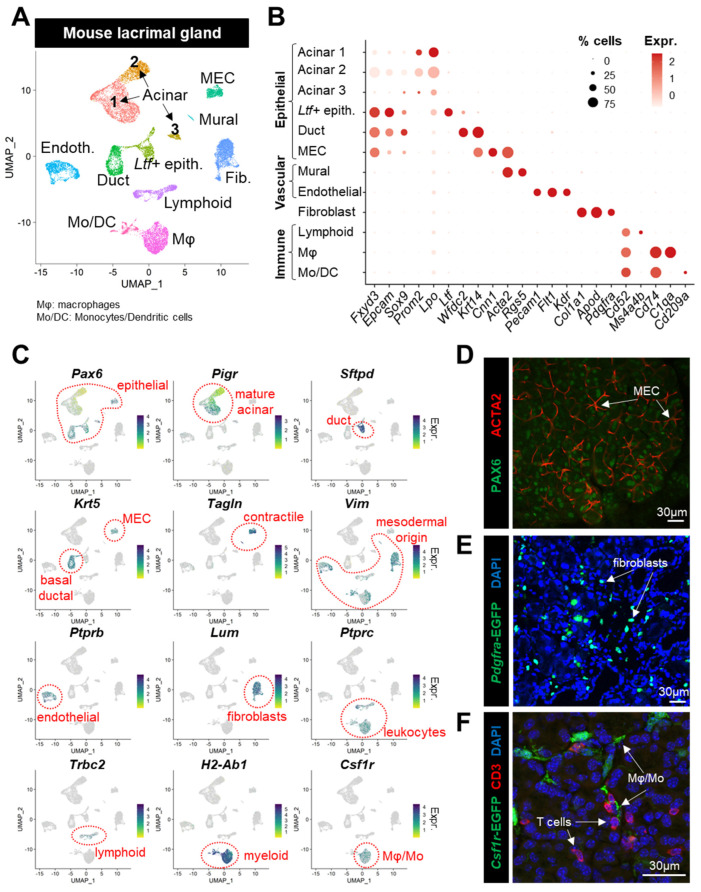
Identification of the main cell types composing the lacrimal gland. (**A**) UMAP embeddings of lacrimal gland (LG) cells from 2-month-old mice following data integration (one female and two male samples, one sample = three mice). (**B**) Expression level of canonical markers used for the annotation of major cell types composing the LG. The size of dots indicates the percentage of cells expressing the corresponding gene, and the color scale correlates with the average expression level (red is high). (**C**) UMAP plot for additional established markers showing the broad distribution of each specific feature within a single cluster or compartment (epithelial, vascular, and immune) circled by dashed red shapes. Cells are colored based on their expression level for the corresponding gene (blue is high). Cells with no expression are shown in grey. (**D**–**F**) Representative confocal images of young (1–3-month-old) mouse LG showing the distribution of (**D**) PAX6, expressed by epithelial cells, and ACTA2, labeling all myoepithelial cells, (**E**) green fluorescent fibroblasts in the *Pdgfra*^EGFP^ reporter mouse, and (**F**) green fluorescent macrophages (Mφ)/monocytes and CD3+ T cells (red) in the *Csf1r*^EGFP^ reporter mouse line.

**Figure 2 cells-12-01435-f002:**
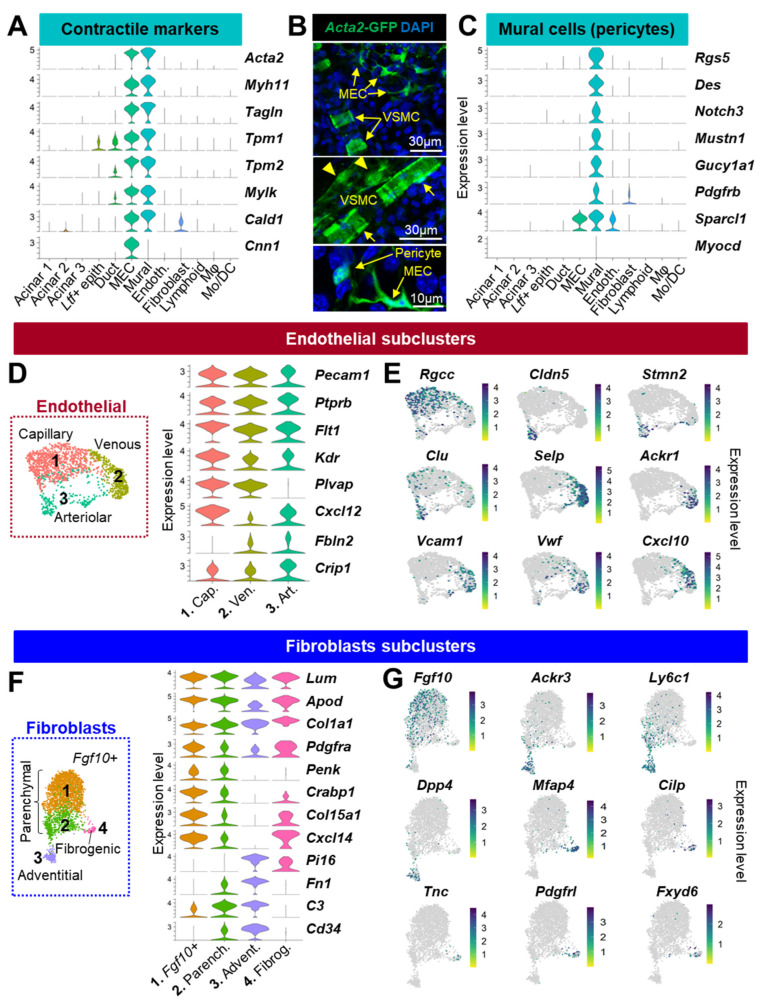
Transcriptional profiles of the stromal cells of the lacrimal gland. (**A**) Violin plots for the expression of contractile genes across the main cell types composing the LG. (**B**) Confocal images of LG whole mounts from the *Acta2*-GFP reporter mouse showing the distinctive morphologies of vascular mural cells and typical star-shaped MECs. Vascular smooth muscle cells (VSMCs) have a different phenotype depending on the vessel they surround: spindle-shaped VSMCs indicate pre-capillary arterioles (arrows), while VSMCs from post-capillary venules have cytoplasmic processes (arrowheads). Pericytes can easily be identified on capillaries due to their prominent nucleus and short processes. Nuclei were visualized with DAPI. (**C**) Normalized expression of the key genes defining pericytes in the present LG dataset. (**D**,**E**) Subclustering evidenced endothelial cells of capillary, venous, and arterial types (see cropped UMAP plot of the endothelial cluster in frame) according to the defining genes they (**D**) commonly or (**E**) specifically express. (**F**,**G**) Fibroblast heterogeneity was evaluated by subclustering (see cropped UMAP plot in frame). (**F**) Shared or (**G**) restricted expression of marker genes evidenced parenchymal (*Fgf10*+ and *Fgf10*-), adventitial, and fibrogenic-like subsets of fibroblasts. Violins are colored by cluster identity.

**Figure 3 cells-12-01435-f003:**
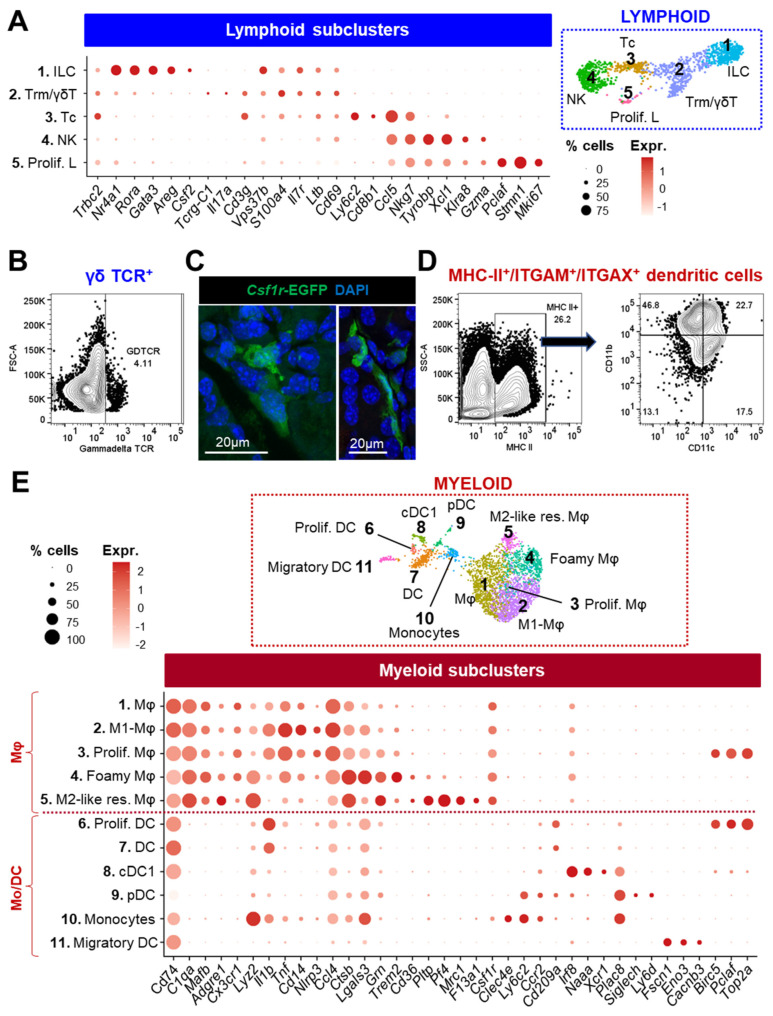
Many different immune cells reside in the steady-state lacrimal gland. (**A**) Expression level of marker genes identifying the lymphoid subclusters (see cropped UMAP to the right). (**B**) Representative plots for the analysis of dissociated LG (3–4-month-old females) by flow cytometry showing the presence of γδ-TCR+ cells indicative of γδ T cells (**C**). Representative confocal images of LG frozen sections from young (1–2-month-old) *Csf1r*-EGFP mice illustrating GFP+ macrophages (Mφ) of various morphologies. Nuclei were stained with DAPI. (**D**) Detection of MHC-II+/ITGAM+/ITGAX+ dendritic cells (DC) by flow cytometry analysis of dissociated LG. (**E**) Subclustering analysis of the myeloid populations (see cropped UMAP plot in frame) identified 11 subtypes of Mφ and monocytes (Mo)/DC characterized by relevant marker genes shown on the dot plot. On all gene expression plots, the size of dots indicates the percentage of cells expressing the corresponding gene, and the color scale correlates with the average expression level (red is high).

**Figure 4 cells-12-01435-f004:**
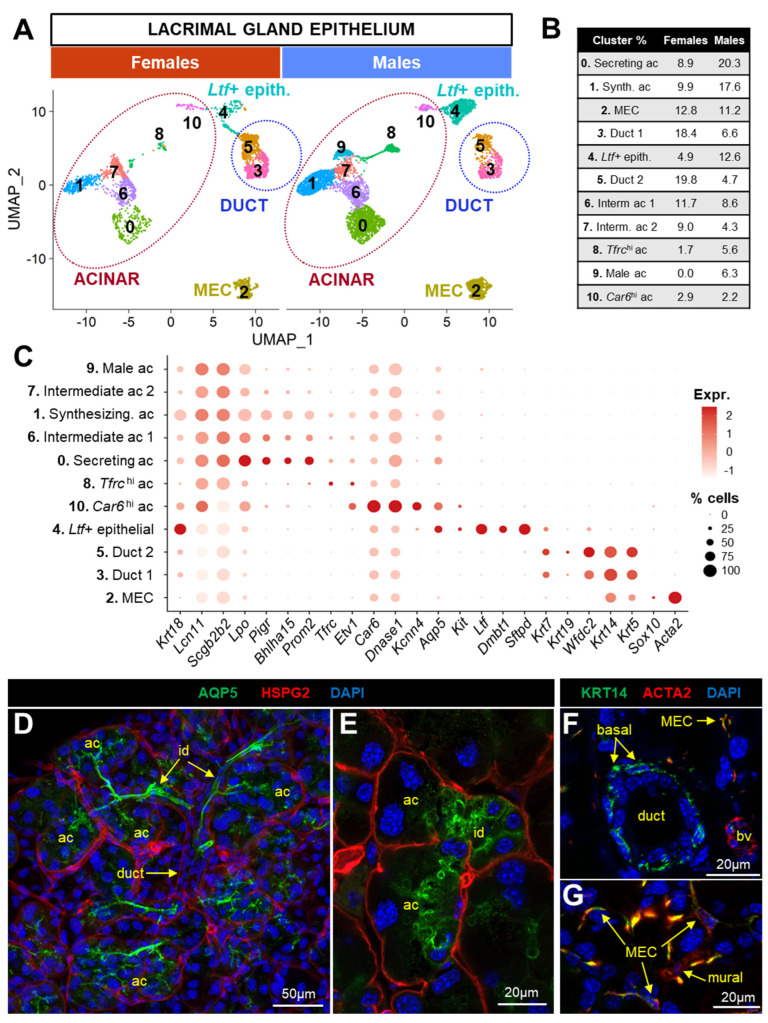
Unexpected complexity of the lacrimal gland epithelium. The epithelial fraction of the LG dataset was subjected to a separate analysis to reveal epithelial subpopulations of the young LG. (**A**) UMAP embeddings of epithelial clusters were split into two plots to visualize sex differences in cell composition. (**B**) The proportion of cells in each epithelial cluster depending on sex. (**C**) The expression of marker genes defining epithelial clusters. Dot size indicates the percentage of cells expressing the corresponding gene, and the color scale correlates with the average expression level (red is high). (**D**–**G**) Representative confocal images of (**D**,**E**) LG whole mounts stained for heparan sulfate proteoglycan (HSPG2, red) and aquaporin-5 (AQP5, green). HSPG2 labeled the basement membrane of acini, ducts, and blood vessels. AQP5 is found at the apical part of the acini and small/intercalated ducts but is absent in the large multilayered ducts (see (**D**), projection of 6 µm thick z-stack). (**F**,**G**) LG frozen sections from 2–6-month-old mice immunostained for ACTA2 (to visualize MECs and mural cells wrapped around blood vessels) and KRT14, which is specifically expressed by (**F**) ACTA2- basal ductal cells and (**G**) ACTA2+ MECs but not vascular cells. Nuclei were counterstained with DAPI.

**Figure 5 cells-12-01435-f005:**
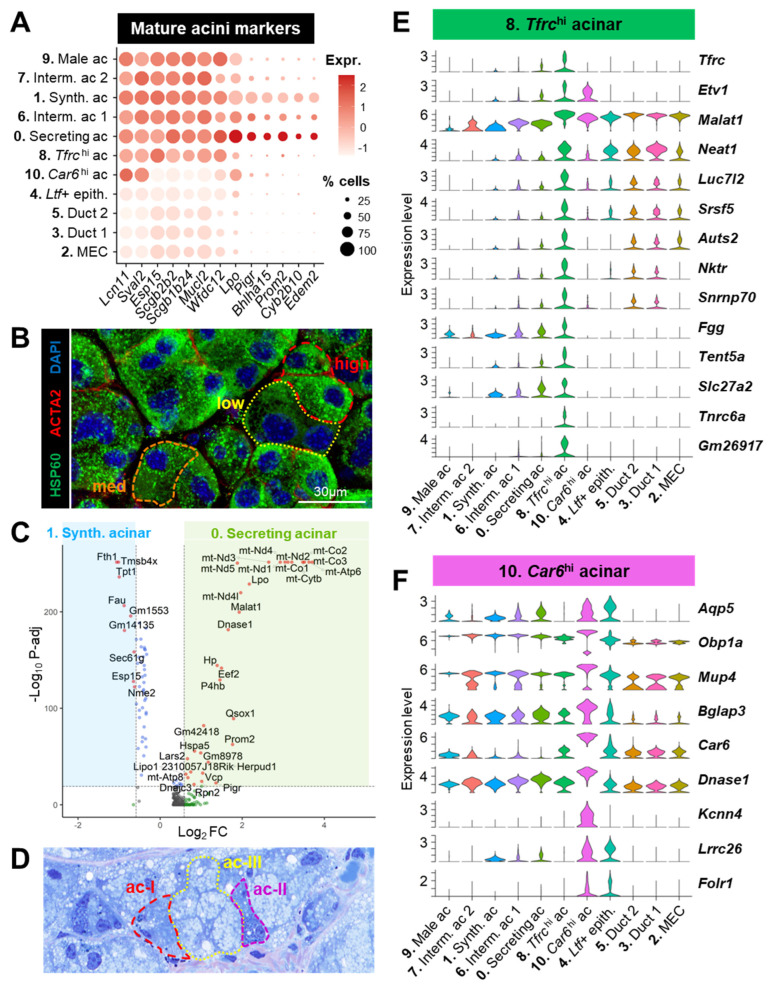
Intrinsic heterogeneity of the acinar compartment of the lacrimal gland. (**A**) Expression level of genes commonly found in the acinar compartment and features associated with mature fully differentiated acinar phenotypes. Dot size indicates the percentage of cells expressing the corresponding gene, and the color scale correlates with the average expression level (red is high). (**B**) Representative confocal images of LG whole mounts from 5–6-month-old mice immunostained for ACTA2 and the mitochondrial marker HSP60 (green). The mitochondrial content is heterogenous across acinar cells (surrounded by ACTA2+ MECs) as shown by the low (yellow dashes), medium (orange dashes), and high (red dashes) staining of acinar cytoplasm. (**C**) Volcano plot for differentially expressed genes between acinar cluster #0 and #1. To improve readability, all *Rps* and *Rpl* genes coding for ribosomal proteins enriched in cluster #1 were omitted from the plot. Only genes expressed in at least 20% of the cells of either of the two clusters were considered. Dashed lines intersect the x-axis at fold change = ±1.5 and the y-axis at *p*-adj = 10^−20^. Positive fold changes indicate an upregulation in cluster #0 compared to cluster #1. (**D**) One-micron section of 2-month-old LG stained with toluidine blue reveals acinar cells containing a low number of granules (ac-I), some cytoplasmic granules that may contain glycoproteins as suggested by the dark blue staining (ac-II), and large acinar cells densely filled with light blue granules (ac-III). (**E**,**F**) Violin plots for the expression of marker genes defining (**E**) acinar cluster #8 and (**F**) acinar cluster #10. Violins are colored by cluster identity.

**Figure 6 cells-12-01435-f006:**
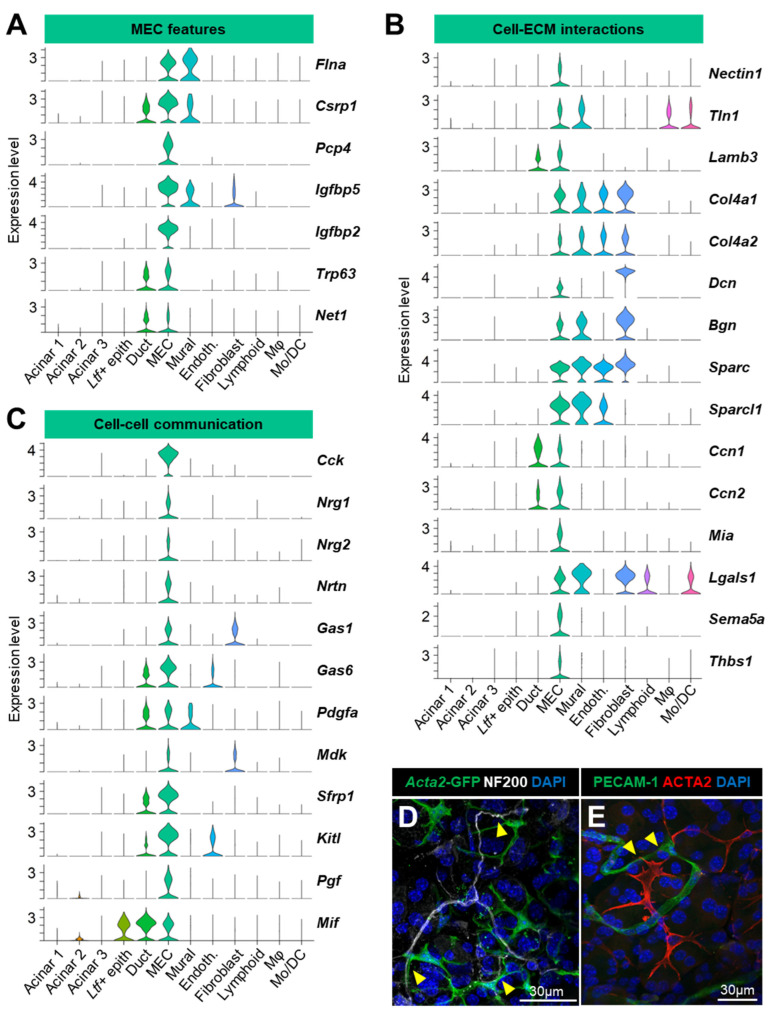
Myoepithelial cells (MEC) have the transcriptional repertoire to interact and communicate with other cell types. (**A**–**C**) The expression level in the whole LG dataset for (**A**) markers defining MECs, genes coding for (**B**) cell–matrix interaction molecules, and (**C**) cell–cell communication factors, significantly enriched in MECs compared to other cell types composing the LG. (**D**,**E**) Projection of z-stacks from LG whole mounts stained for (**D**) the neuronal marker Neurofilament 200 (NF200, white) in the *Acta2*^GFP^ reporter mouse and (E) the endothelial marker PECAM-1 (green) and MEC marker ACTA2 (red) in a wildtype mouse. Yellow arrowheads indicate physical interactions between MECs and (**D**) neuronal or (**E**) endothelial cells, respectively.

**Figure 7 cells-12-01435-f007:**
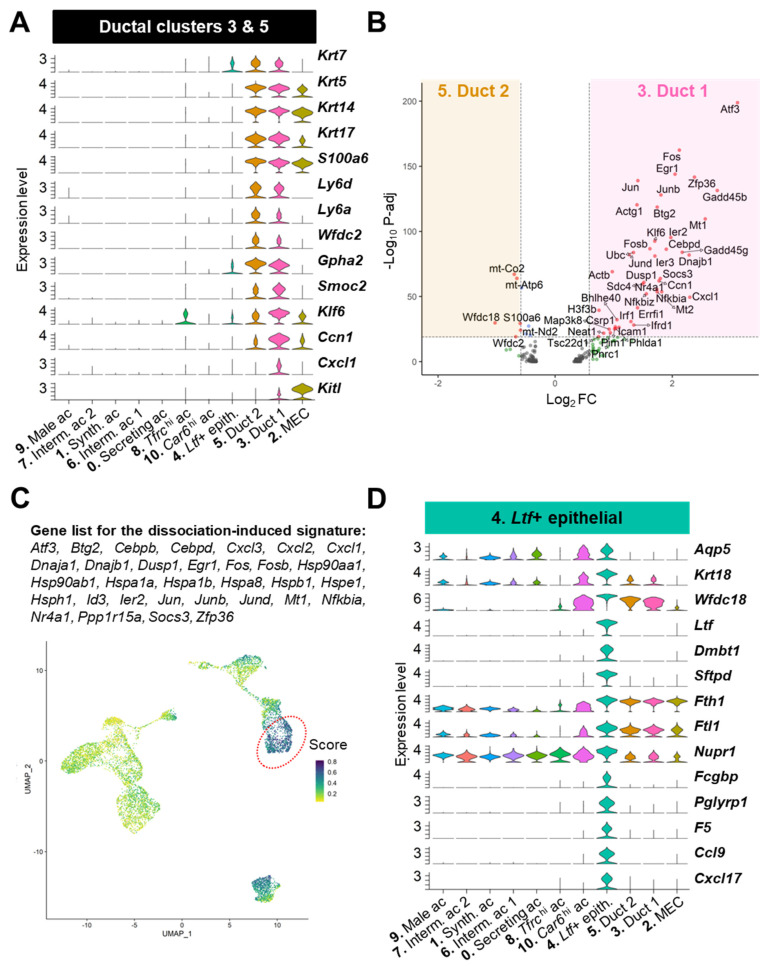
Transcriptional profiling of the ductal population of the lacrimal gland. (**A**) Expression level of markers for the ductal clusters #3 and 5. (**B**) Volcano plot for the differentially expressed genes between ductal cluster #3 and #5. Only genes expressed in at least 20% of the cells of either of the two clusters were considered. Dashed lines intersect the x-axis at fold-change = ±1.5 and the y-axis at *p*-adj = 10^−20^. Positive fold changes indicate an upregulation in cluster #3 compared to cluster #5. (**C**) UMAP plot showing the cell scores for the previously published dissociation-induced gene signature. Higher scores (blue) were found in cluster #3. (**D**) Expression of genes defining cluster #4. Violins are colored by cluster identity.

**Figure 8 cells-12-01435-f008:**
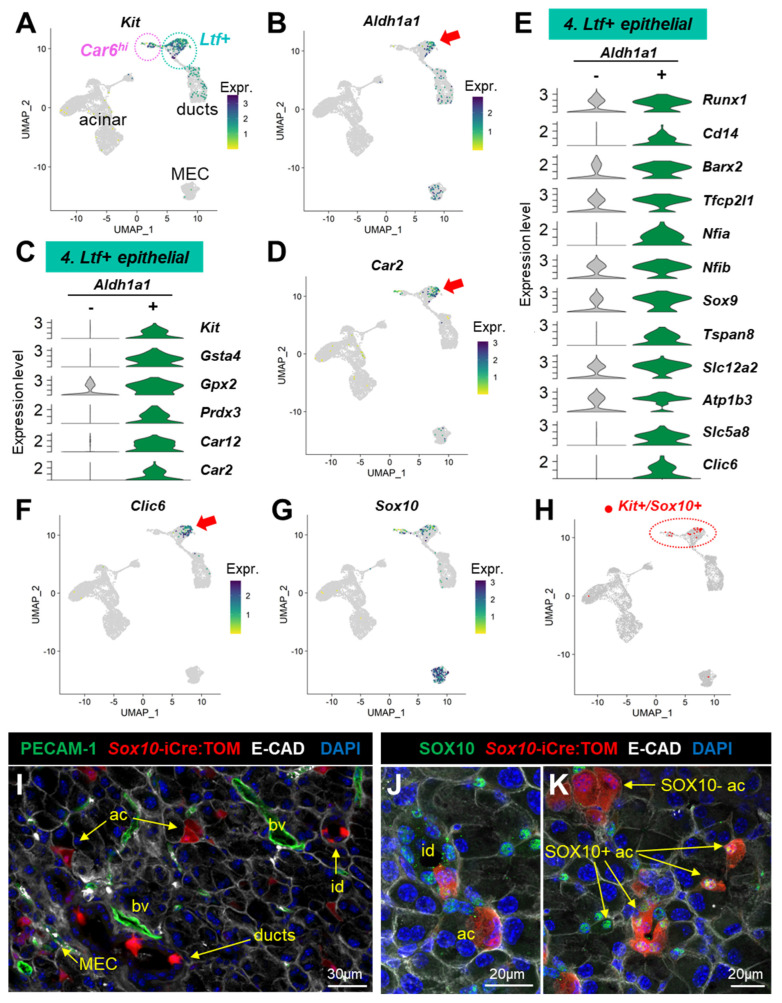
Identification of distinct *Kit*+ progenitor subsets in the lacrimal gland epithelium. (**A**,**B**) UMAP plots identify (**A**) *Kit*+ and (**B**) *Aldh1a1*+ expressing cells in the *Ltf*+ epithelial cluster (red arrow). (**C**) Expression level of genes enriched in the *Aldh1a1*+ cells of the *Ltf*+ cluster. (**D**) UMAP plot for *Car2* expression in the LG epithelium. (**E**) Expression level of genes enriched in the *Aldh1a1*+ cells of the *Ltf*+ cluster related to progenitor-like properties and ion transport. (**F**,**G**) UMAP plot for (**F**) *Clic6* and (**G**) *Sox10* expression in LG epithelium. (**H**) *Kit*+/*Sox10*+ cells of the LG epithelium are highlighted in red and concentrate in the *Car6^hi^* acinar and *Ltf+* ductal clusters (circled area). (**I**–**K**) Two-month-old *Sox10*-iCre:TOM mice received TM for two days and were sacrificed (**I**) three days or (**J**,**K**) one week after the first TM injection. The LG was harvested and stained for (**I**) PECAM-1 (blood vessels (bv) marker, green) and E-cadherin (E-CAD, epithelial marker, white), or (**J**,**K**) SOX10 (green), or E-CAD (white). TOM+ cells (red) derived from *Sox10*-expressing cells were identified as MECs, acinar (ac), ductal, and intercalated duct (id) cells. Most of the MECs were SOX10+. Many SOX10+ cells were found in intercalated ducts and at the junction with acini. SOX10+ acinar-like cells are smaller and found centrally in the vicinity of the acini lumen, while the TOM+/SOX10- acinar cells form clusters of large cells.

**Figure 9 cells-12-01435-f009:**
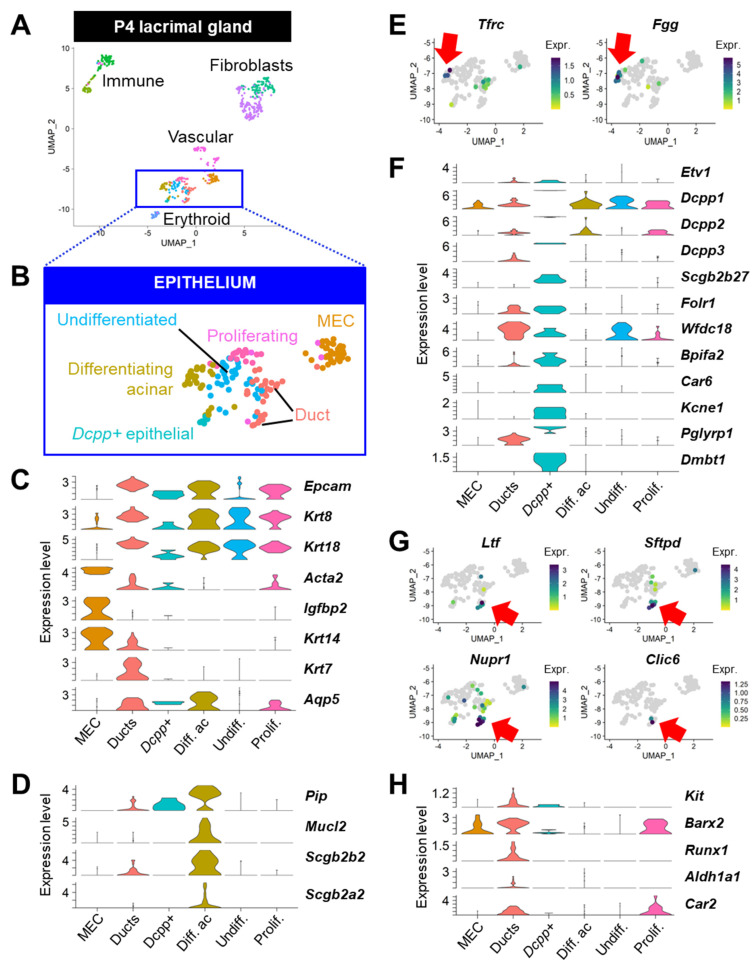
The developing epithelium of the lacrimal gland shows key features of immature adult subsets. (**A**,**B**) UMAP embeddings of the (**A**) LG dataset analyzed at postnatal day 4 (P4) and including (**B**) six epithelial subclusters. (**C**,**D**) Normalized expression level of genes defining (**C**) the main cellular lineages and (**D**) differentiating the acinar cells. (**E**) UMAP plot showing *Tfrc/Fgg*^hi^ differentiating acinar cells. (**F**) Normalized expression in the *Dcpp*+ cluster of marker genes and features of the adult *Car6*^hi^ acinar cluster. (**G**) Some of the main characteristic genes of the adult *Ltf*+ cluster are found in a subset of developing ducts. (**H**) The developing ducts are enriched in the expression of genes specific to *Aldh1a1*+ cells of the centroacinar/terminal duct (CA/TD)-like cells of the adult *Ltf*+ epithelial cluster.

**Figure 10 cells-12-01435-f010:**
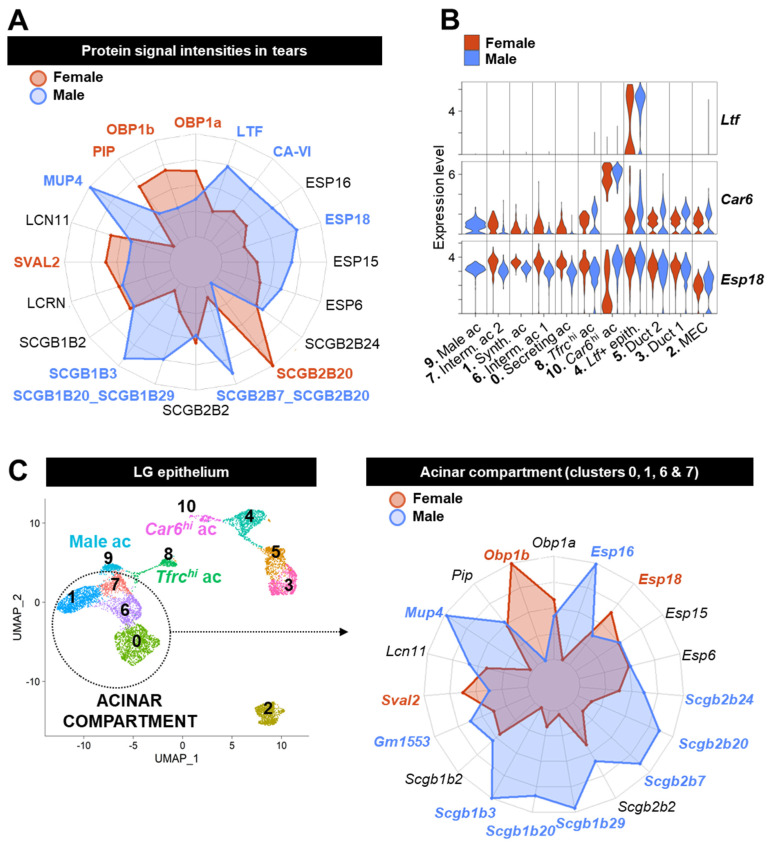
Sex dimorphism in tear composition mainly originates from the acinar compartment. (**A**) Spider chart summarizing the average level of tear components at the protein level in mouse tears (n = 6). (**B**) Normalized expression levels of abundant tear components highly expressed by clusters that do not belong to the main acinar compartment. Red and blue violins correspond to female and male data, respectively. (**C**) Spider chart summarizing the average level of tear components at the mRNA level in the acinar compartment of the LG epithelium (including clusters #0, 1, 6, and 7 as shown on the plot to the left). (**A**,**C**) Webs are colored in blue or red depending on whether they belong to male or female data, respectively. For each protein/transcript, the maximum of the chart corresponds to the sum of the respective averages in male and female mice so that strictly sex-specific tear products hit the largest polygon, while equally expressed proteins/genes colocalize in the middle of the corresponding axis. Sex-biased tear products are indicated in bold and colored in blue or red if they are significantly enriched (fold change > 1.5; *p*-adj < 0.05) in males or females, respectively.

## Data Availability

All data are available in the main text or the Appendix A. The R file used to analyze the previously published sequencing data of the postnatal lacrimal gland (P4) was found on Panglodb.se under identifier SRS2290358. Sequencing data for the scRNAseq analysis of the lacrimal glands from two-month-old mice were deposited in the Gene Expression Omnibus (GEO) database (www.ncbi.nlm.nih.gov/geo, accessed on 15 May 2023) under accession #GSE232146. R scripts and Seurat objects used for this study were uploaded on Zenodo (zenodo.org, accessed on 15 May 2023) with https://doi.org/10.5281/zenodo.7927055, accessed on 15 May 2023.

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
