# Peer review of "The First Transcriptomic Atlas of the Adult Lacrimal Gland Reveals Epithelial Complexity and Identifies Novel Progenitor Cells in Mice"

_cells, 2023, doi:10.3390/cells12101435_

Round 1

Reviewer 1 Report

General Comments

Here the authors explore in exquisite single cell detail the mRNA expression of the exorbital lacrimal gland in female and male mice.  This tour de force exploration of all cell types, as largely backed up by immunostaining and cell lineage tracing is very impressive. 

Other comments:

A. Methods

-       What is the ~% cell yield from tissue to cells for scRNAseq?  90 min of tissue digestion seems very long versus acinar cell isolation for cell culture.

-       The authors do not appear to cite 'minSCe' and its guidelines, as suggested by Fullgrabe et al, Nat Biotech '19.  minSCe is a 'minimum set of single-cell metadata categories and a checklist of information that can be used to describe a single-cell assay in sufficient detail to enable analysis of the transcriptomic data'. 

-       Why were female and male samples processed by different institutions?  Could this introduce variability?

-       Line 162: Please clarify in the Methods how many animals comprise ‘one female and two male samples…’.  Earlier it is stated that each scRNAseq sample consists of six lacrimal glands from three mice aged 2 months old.

-       What is the number of animals used for immunostaining and what immunostaining negative control(s) was used?

-       Lines 224 – 243: The Sigma catalogue for MAB1948P indicates that ‘The antibody recognizes a high molecular weight core protein of Heparan Sulfate Proteoglycan (Perlecan)’.  The gene symbol is Hspg2.  Rather than anti-HS as an abbreviation that implies selectivity for heparan sulfate chains, it would be best to use ‘anti-Hspg2’.  It would be similarly best to refer to the antigen of all other antibodies using ‘anti-gene symbol’.  For example, anti-CD31 should be ‘anti-Pecam1’, anti-a-smooth muscle actin should be ‘anti-Acta1’, etc.  This aligns with use of gene symbols in the Results and will avoid nomenclature confusion into the future.  Please specify catalog numbers for secondary antibodies.

-       How many replicate scRNA seq and other experiments were performed?

-       Lines 247 – 254: ‘Half-thin sections’ is an odd term.  Why not replace with ‘one micron sections’.

-       Lines 256 – 266: Use gene symbols for reference to antigens, ie. anti-CD45 should be ‘anti-Ptprc’. As noted, this aligns with use of gene symbols in the Results and will avoid nomenclature confusion into the future.  

B. Results

- In each of the UMAP clusters, what % of the data is being clustered?

- In Fig. 4B, how statistically significant are the female/male differences in UMAP clusters?

- With the exception of Tfrc, Folr1, Kit, Cd36, Gpr108 and Pigr, no other receptors are apparently mentioned (perhaps there are others in the Supplemental data).  Admittedly expression levels of receptors would be expected to be low, but some should figure prominently.  Is this indeed a reflection of the relatively low 20,000 reads per cell?

- Lines 784, 785: a citation appears to be lacking for 'a previously published scRNA-seq data...';   'a' in this sentence is unnecessary.

C. Discussion

- Lines 1098 - 1100: The authors discuss differential cell loss during single cell isolation.  Could the authors offer any insight into this?  Are the cell types that were underrepresented also so in other scRNAseq studies in other tissues?

Author Response

General Comments

Here the authors explore in exquisite single cell detail the mRNA expression of the exorbital lacrimal gland in female and male mice.  This tour de force exploration of all cell types, as largely backed up by immunostaining and cell lineage tracing is very impressive. 

We express our gratitude to the reviewer for his/her gracious comments. Although the lacrimal gland is a remarkable model for the study of exocrine glands, it is often under-studied when compared to other eye-related organs. We are delighted to learn that our work was well-received and that our findings and data will be shared with the scientific community.

Other comments:

  1. Methods

-       What is the ~% cell yield from tissue to cells for scRNAseq?  90 min of tissue digestion seems very long versus acinar cell isolation for cell culture.

We acknowledge the reviewer's suggestion that a shorter digestion time could be beneficial for maintaining cell integrity. However, our experimentation with mice older than one month showed that 45-60 minutes of digestion did not significantly improve cell viability, and failed to produce single cells from the epithelial compartment. Although longer digestion times (120 minutes) slightly decreased cell viability, they did not yield a significant increase in cell production. Unfortunately, cold-active enzymes, as suggested by previously published studies (O'Flanagan et al., 2019) failed to disintegrate the extracellular matrix of adult mouse LGs. Therefore, after considering all these factors, we determined that a 90-minute digestion period at 37C was the best compromise between epithelial single cell yield and cell viability.

The tissue dissociation protocol typically generates between 2-8 million cells, with male LGs yielding substantially higher cell counts due to their larger size and being used to establish the dissociation protocol. However, the dead cell removal step is associated with significant cell loss, with a considerable number of cells becoming trapped in the columns. Following dead cell removal, the female LGs yielded between 40K to 100K cells, while the male LGs produced around 200K cells. Although these numbers may seem low compared to the total cell count after dissociation, they are adequate for scRNAseq analysis, given that only 15K cells were loaded for each sample, as recommended by the 10X protocol.

-       The authors do not appear to cite 'minSCe' and its guidelines, as suggested by Fullgrabe et al, Nat Biotech '19.  minSCe is a 'minimum set of single-cell metadata categories and a checklist of information that can be used to describe a single-cell assay in sufficient detail to enable analysis of the transcriptomic data'. 

We have deposited our scRNAseq data on the GEO database under the accession number GSE232146. All technical details that conform to the minSCe guidelines will be provided on each sample's GEO webpage. To ensure complete transparency about the parameters and methods used in our analysis, our R scripts and R objects used to generate the figures will be accessible upon publication from Zenodo at doi: 10.5281/zenodo.7927055.

-       Why were female and male samples processed by different institutions?  Could this introduce variability?

Females were processed at Scripps, which allowed us to obtain a pilot grant to analyze sex dimorphism. However, this funding could only be used for services provided by Salk core facilities, which is located very close to Scripps. To minimize technical variability, lacrimal glands from both males and females were harvested at the same time of day and processed using the exact same dissociation protocol and enzyme batches by the same people. The scRNAseq analysis was performed using the same 10X kits and protocols for male and female samples. Subsequently, the male and female samples were integrated by CCA to minimize batch-effects and enable cluster alignment despite sex differences. Our experience indicates that even samples loaded onto the same microfluidic chip but in different wells usually require integration.

-       Line 162: Please clarify in the Methods how many animals comprise ‘one female and two male samples…’.  Earlier it is stated that each scRNAseq sample consists of six lacrimal glands from three mice aged 2 months old.

We apologize for any confusion. We would like to clarify that each sample used in our study consisted of six LGs, which comprised of three (2 months old) animals. For the final analysis presented in this manuscript, we retained one female sample (n=3 mice) and two male samples (n=6 mice). Therefore, our single cell atlas reflects the LG composition of 9 animals in total. These details were added to the Methods at lines 154, 164-165.

-       What is the number of animals used for immunostaining and what immunostaining negative control(s) was used?

We used a minimum of 4 mice for immunostaining, including both male and female animals in our investigation. The images included in our manuscript reflect our observations in both sexes. To verify the specificity of the signals, we omitted the primary antibody during the incubation with the secondary antibody. We have added these details in the Method section of our manuscript.

For lineage tracing to confirm that there was no spontaneous Cre recombination in Sox10-Cre-TOM mice, we analyzed uninjected littermates and injected tdTOMfl littermates. We have added these details to the Methods.

-       Lines 224 – 243: The Sigma catalogue for MAB1948P indicates that ‘The antibody recognizes a high molecular weight core protein of Heparan Sulfate Proteoglycan (Perlecan)’.  The gene symbol is Hspg2.  Rather than anti-HS as an abbreviation that implies selectivity for heparan sulfate chains, it would be best to use ‘anti-Hspg2’.  It would be similarly best to refer to the antigen of all other antibodies using ‘anti-gene symbol’.  For example, anti-CD31 should be ‘anti-Pecam1’, anti-a-smooth muscle actin should be ‘anti-Acta1’, etc.  This aligns with use of gene symbols in the Results and will avoid nomenclature confusion into the future.  Please specify catalog numbers for secondary antibodies.

In agreement with the reviewer’s comments, we corrected “anti-HS’ by “anti-HSPG2” in the methods and on figures where appropriate. To improve clarity, we changed protein names to their alternate names if they matched the official gene name (eg CD31 was replaced by PECAM-1) and specified their usual synonyms in the text. For protein names that do not match the gene symbol (eg ECAD for Cdh1), we added the gene name in the text.

-       How many replicate scRNA seq and other experiments were performed?

We conducted scRNA-seq experiments on two separate days, with one day dedicated to female samples and the other day for male samples. Other results reflect at least 3 independent experiments performed on males and females. However, in the flow cytometry analysis, we only used female mice, since the purpose of this experiment was to validate the presence of specific cell types in the LGs. We have included these details to accurately describe our experimental methodology in the relevant sections of our manuscript.

-       Lines 247 – 254: ‘Half-thin sections’ is an odd term.  Why not replace with ‘one micron sections’.

Thank you for pointing out this mistake, “half-thin section” term has been replaced with ‘one micron section’.

-       Lines 256 – 266: Use gene symbols for reference to antigens, ie. anti-CD45 should be ‘anti-Ptprc’. As noted, this aligns with use of gene symbols in the Results and will avoid nomenclature confusion into the future.  

We changed the protein names according to the reviewer’s suggestion.

  1. Results

- In each of the UMAP clusters, what % of the data is being clustered?     

Starting from Cell-Ranger-prefiltered matrices, the numbers of cells that passed our sequential QC steps are shown for each sample in the table below. On average, 70% of the CellRanger output was used for the final clustering shown in the manuscript.

Table: Cell numbers at the different steps of data processing

Sample

# cells in the filtered matrix of CellRanger output

# cells after removing cells with high decontX score and doublets

# cells with at least 500 nUMI, 200 genes and less than 15% mitochondrial transcripts

Female 1

9737

8250

7056

Male 1

6422

5138

4723

Male 2

6899

4799

4493

The distribution of cell clusters can be found on Fig. S2C for the main populations and on Fig. 4B for the epithelium sub-clustering.

- In Fig. 4B, how statistically significant are the female/male differences in UMAP clusters?

As we only have one female sample and two male samples, it is not feasible to perform statistical tests on this variable. Furthermore, variations in cell composition should be interpreted with care, particularly after tissue dissociation. Therefore, we have only conducted a qualitative evaluation of any notable differences, such as the absence of cluster #9 in females. We acknowledge that a quantitative assessment of cell proportions would require a larger sample size and histological validations, if possible. Nonetheless, we have made every effort to provide a thorough and accurate interpretation of our results, taking into consideration the limitations of scRNAseq studies.

- With the exception of Tfrc, Folr1, Kit, Cd36, Gpr108 and Pigr, no other receptors are apparently mentioned (perhaps there are others in the Supplemental data).  Admittedly expression levels of receptors would be expected to be low, but some should figure prominently.  Is this indeed a reflection of the relatively low 20,000 reads per cell?

We indeed showed additional receptors (eg Flt1, Kdr, Pdgfrb, Pdgfra, Cd209a, Cd74, Csf1r, Trem2, Ackr1, Cd3g, Cd3d, Il7r, Trac, Klra4, Cd14, Mrc1, Cd7, Nktr). We have included some of these receptors in the main text, while others are provided in the supplemental materials. In our study we mainly focused on highlighting the features that best characterize the cell clusters or specific functions, such as cell-cell communication factors in MECs.  

In the violin plot below, we have included examples of other receptors that we detected across our dataset. For instance, we have presented the normalized expression levels for receptors that we and others have previously reported to be expressed by LG epithelial cells, such as Egfr, Fgfr1, and P2rx4. Our analysis also revealed that the sequencing depth attained by scRNAseq did not enable the significant detection of some other known receptors, as only a small number of epithelial cells and MECs were found to be positive for Fgfr2 and Oxtr, respectively.

- Lines 784, 785: a citation appears to be lacking for 'a previously published scRNA-seq data...';   'a' in this sentence is unnecessary.

Thank you for pointing out this mistake, it has been corrected.

  1. Discussion

- Lines 1098 - 1100: The authors discuss differential cell loss during single cell isolation.  Could the authors offer any insight into this?  Are the cell types that were underrepresented also so in other scRNAseq studies in other tissues?

The mouse submandibular gland is the most similar tissue for which comprehensive scRNAseq data are available, as demonstrated in studies conducted by Hauser et al (2020) and Horeth et al (2021). However, these studies revealed an underrepresentation of epithelial cells, especially acinar cells in comparison to the intact tissue. Similarly, our dataset and datasets of others also failed to collect neurons due to their size and long processes. It is likely that the harsh conditions required for epithelial duct dissociation are not very well compatible with the preservation of large and fragile cell types such as acinar and neurons. Moreover, populations present in relatively low numbers would require larger datasets to form significant and detectable cell clusters. To address these challenges, it is essential to develop specialized protocols that are tailored to the specific cell types of interest. If markers are known, rare cell populations can be enriched using FACS. However, some cell types may not survive well under sorter-induced cell stress, limiting their usefulness in such experiments.

Reviewer 2 Report

Congratulations on the great work and exploring scRNAseq to investigate the cellular complexity. 

Author Response

We would like to express our appreciation to the reviewer for  his/her kind remarks.